# Behaviour-based dependency networks between places shape urban economic resilience

**Takahiro Yabe** [1,2,3] ✉, **Bernardo García Bulle Bueno**[1], **Morgan R. Frank** [4,5,6], **Alex Pentland** [1,6] **& Esteban Moro** [1,6,7] ✉

Disruptions, such as closures of businesses during pandemics, not only affect businesses and amenities directly but also influence how people move, spreading the impact to other businesses and increasing the overall economic shock. However, it is unclear how much businesses depend on each other during disruptions. Leveraging human mobility data and same-day visits in five US cities, we quantify dependencies between points of interest encompassing businesses, stores and amenities. We find that dependency networks computed from human mobility exhibit significantly higher rates of long-distance connections and biases towards specific pairs of point-of-interest categories. We show that using behaviour-based dependency relationships improves the predictability of business resilience during shocks by around 40% compared with distance-based models, and that neglecting behaviour-based dependencies can lead to underestimation of the spatial cascades of disruptions. Our findings underscore the importance of measuring complex relationships in patterns of human mobility to foster urban economic resilience to shocks.

Cities are central drivers of economic growth, owing to their agglomeration of businesses and amenities connected through dense social interactions, financial transactions and human activities[1]. The high level of complexity enables ideas, innovations and information to spread across organizations and communities[2,3]. At the same time, high urban connectivity allows shocks to cascade across time, space and system components. Examples of urban networks include labour markets[4], global supply chains[5,6], networks of social encounters[7] and infrastructure networks[8]. Understanding the spread of shocks across urban spaces, among businesses and urban amenities, is crucial for resilient urban planning policies, which aim to mitigate disruptions and to improve the recovery speed and quality of businesses and organizations in cities[9].

From the global[10,11] to the regional[12] scale, studies of the economic resilience of industries focus on predicting the interindustry, cascading along connections in the supply chain networks[5,13–16]. However, similar to supply chains, the demand side also has a network of dependencies, mediated by human mobility, that can curb or amplify shocks caused by changes in consumer behaviour. For example, sustained remote work during the coronavirus disease 2019 (COVID-19) pandemic[17,18] has been associated with decreased foot traffic to cafes near central business districts[19], and surveys have indicated that patronage to office-district restaurants is likely to decrease[20]. Such dependencies between industries could cause shocks to cascade, thus posing a substantial threat to the economic resilience of business ecosystems and their corresponding development and design[21].

[1]Institute for Data, Systems, and Society, Massachusetts Institute of Technology, Cambridge, MA, USA. [2]Center for Urban Science and Progress (CUSP), Tandon School of Engineering, New York University, Brooklyn, NY, USA. [3]Department of Technology Management and Innovation, Tandon School of Engineering, New York University, Brooklyn, NY, USA. [4]Department of Informatics and Networked Systems, University of Pittsburgh, Pittsburgh, PA, USA. [5]Digital Economy Lab, Institute for Human-Centered AI, Stanford University, Stanford, CA, USA. [6]Media Lab, Massachusetts Institute of Technology, Cambridge, MA, USA. [7]Network Science Institute, Department of Physics, Northeastern University, Boston, MA, USA. ✉e-mail: takahiroyabe@nyu.edu; esteban.moroegido@gmail.com

The household production theory, which holds that households allocate resources (time and money) to maximize utility, provides the theoretical basis for our study on patronage behaviour across multiple stores[22]. Empirical studies on consumer behaviour have identified a multitude of factors that affect patronage to multiple stores, including customers' sociodemographic characteristics, store characteristics, available transportation modes and the built environment[23–25]. Despite its importance on characterizing economic resilience, researchers have only recently started to examine general patterns of store patronage[26]. Due to lack of large-scale evidence of mobility and behaviour patterns across stores, dependencies among businesses are typically measured by physical proximity, assuming similar patronage to nearby businesses. As a result, several studies have investigated the resilience of business areas based only on the type and diversity of businesses and amenities[27–29]. These studies fail to incorporate the actual patterns of how individuals visit and interact with different businesses and places.

Recently, studies have used large-scale human mobility data (for example, mobile phone global positioning systems (GPS))[30,31] as scalable low-cost proxies for visitation patterns to various places in urban environments to study behavioural segregation[32], pandemic response[33] and disaster recovery[34]. Moreover, mobile phone location data have been used to estimate the losses of businesses that rely on foot traffic (for example, restaurants and cafes) during disaster events[35,36]. Analysis of human behaviour patterns using mobility and credit card purchasing data have revealed that activity patterns are clustered into a mixture of behavioural lifestyles (for example, health and exercise, local trips, shopping weekends and so on)[37,38], suggesting that certain industries or place categories could have a high dependency on visitors arriving from other specific industries or places. For example, a gas station on a commuting route to business districts might be affected, as well as the cafe in that business district, if people change their visitation patterns to offices. However, studies on economic resilience have so far neglected such interdependent relationships that human behaviour patterns may generate between businesses and other amenities.

This study investigates the interdependencies between urban businesses and amenities using human mobility data and presents a quantitative framework to measuring economic resilience to large-scale behavioural changes. Using a large and longitudinal dataset of GPS location records in five major metropolitan areas in the United States (New York, Boston, Los Angeles, Seattle and Dallas), we construct and analyse behaviour-based dependency networks between businesses and amenities and further use the empirical networks to analyse and simulate the cascades of urban shocks. We find that empirical dependency networks generated via movements between points of interest (POIs) contain a significantly higher rate of long-distance connections between places and are biased towards specific POI category pairs in comparison with a baseline network based on the gravity model. This means shocks in one part of the economy have a greater potential to cascade across a city than would be expected. Analysis reveals that using the behaviour-based dependency network improves the predictability of the resilience of businesses during shocks, such as COVID-19, compared with dependency networks based on physical proximity. We further predict the propagation of changes in visits to POIs under hypothetical external shock scenarios via network simulations and demonstrate how neglecting such dependency relationships may underestimate the extent of the shock. Behaviour-based dependency networks enable better measurements of the effects of urban shocks, including natural hazards, new technology and urban development policies mediated by human behaviour.

## Results

Using a large and longitudinal dataset of GPS location records in five major US metropolitan areas, we construct the behaviour-based dependency network at the level of POIs (for example, businesses and amenities) in cities. Mobility data was provided by Spectus, who supplied anonymous, privacy-enhanced and high-resolution mobile location pings for more than 1 million devices across five US Census core-based statistical areas (CBSAs). All devices within the study opted in to anonymous data collection for research purposes under a general data protection regulation and the California Consumer Privacy Act compliant framework. Our second data source is a collection of over 1 million verified places across five CBSAs, obtained via Safegraph. Within the mobility dataset, we identified stays at places that were detected to be between 10 min and 10 h in duration, and we spatially matched those stays with the closest place locations within 100 m to infer visits to specific POIs. Detailed methods for visit attribution are shown in Supplementary Note 1. We implemented poststratification to ensure the representativeness of the data across regions and income levels (Supplementary Note 1).

In this study, the dependence of a POI $i$ on another POI $j$ is defined as $w_{ij} = n_{ij}/n_i$, where $n_i$ denotes the number of individuals who visit POI $i$ and $n_{ij}$ denotes the number of occurrences that both POIs $i$ and $j$ were visited on the same day, within a 6 h period and visited directly before or after, without any intermediate visits to other POIs (Fig. 1). Because the denominator is based on the number of users who visit the target POI, $w_{ij} \neq w_{ji}$. This simple but intuitive measure considers the asymmetric nature of dependencies between POIs, for example, a cafe could have the majority of the customers coming from a nearby college, but the opposite is rarely the case. Our dependency metric is the simplest way to encode the complex joint probability of visitation patterns to POIs in urban areas; however, it does not capture the causal mechanism of covisitations. By computing the dependency weights $w_{ij} \forall i, j$, we obtain the adjacency matrix of the behaviour-based dependency network $W \in \mathbb{R}^{N \times N}$, where $N$ is the total number of POIs present in the CBSA. The network is weighted and directed with directions indicating dependence (for example, a link from POI $i$ to POI $j$ indicates that $i$ depends on $j$). A bootstrap method was used to compute the standard errors for each $w_{ij}$ and to remove the edges with statistically insignificant weights from the dependency network (Supplementary Note 2). We demonstrate the robustness of the dependency network against the choice of the visit attribution parameters and the choice of the POI dataset in Supplementary Note 3. Furthermore, we show that the network characteristics are not sensitive to the choice of covisit detection parameters, including the time interval between visits, and the number of intermediate stops in the sequence of visits.

### Behaviour-based dependency networks

The behaviour-based dependency networks embed the complex spatial and functional dependencies between businesses and amenities. For each place $i$, we compute the total in-weight $w_i^{in} = \sum_j w_{ji}$ and out-weight $w_i^{out} = \sum_j w_{ij}$, across the five cities. The total in-weight of each place measures to what extent the place is depended by other places in terms of customer visitation patterns, and the out-weight measures how much the place depends its customers on other places. Figure 1a visualizes the total in-weight of all nodes in the behaviour-based dependency network in the Manhattan area of New York, showing the total in-weight $w_i^{in}$ with node sizes, coloured by place categories. Areas such as Times Square, Hudson Yards and the Financial District and places such as the Metropolitan Museum of Art, New York University and Mount Sinai Hospital have a high concentration of in-dependency from other places. To obtain a more qualitative understanding of the dependency network, the network diagram in Fig. 1b shows the average dependencies between POI subcategories in Boston (see Supplementary Note 2 for other cities). Each node represents a POI subcategory (there are 96 of them in the dataset), and the three largest outgoing dependency edges are shown for each node. Node sizes show the in-degree of the constructed network (that is, how many other POI categories depend on that node). Many shopping subcategories including supercentres, department stores, malls and clothing stores, and colleges, cafes and restaurants are depended by many other subcategories.

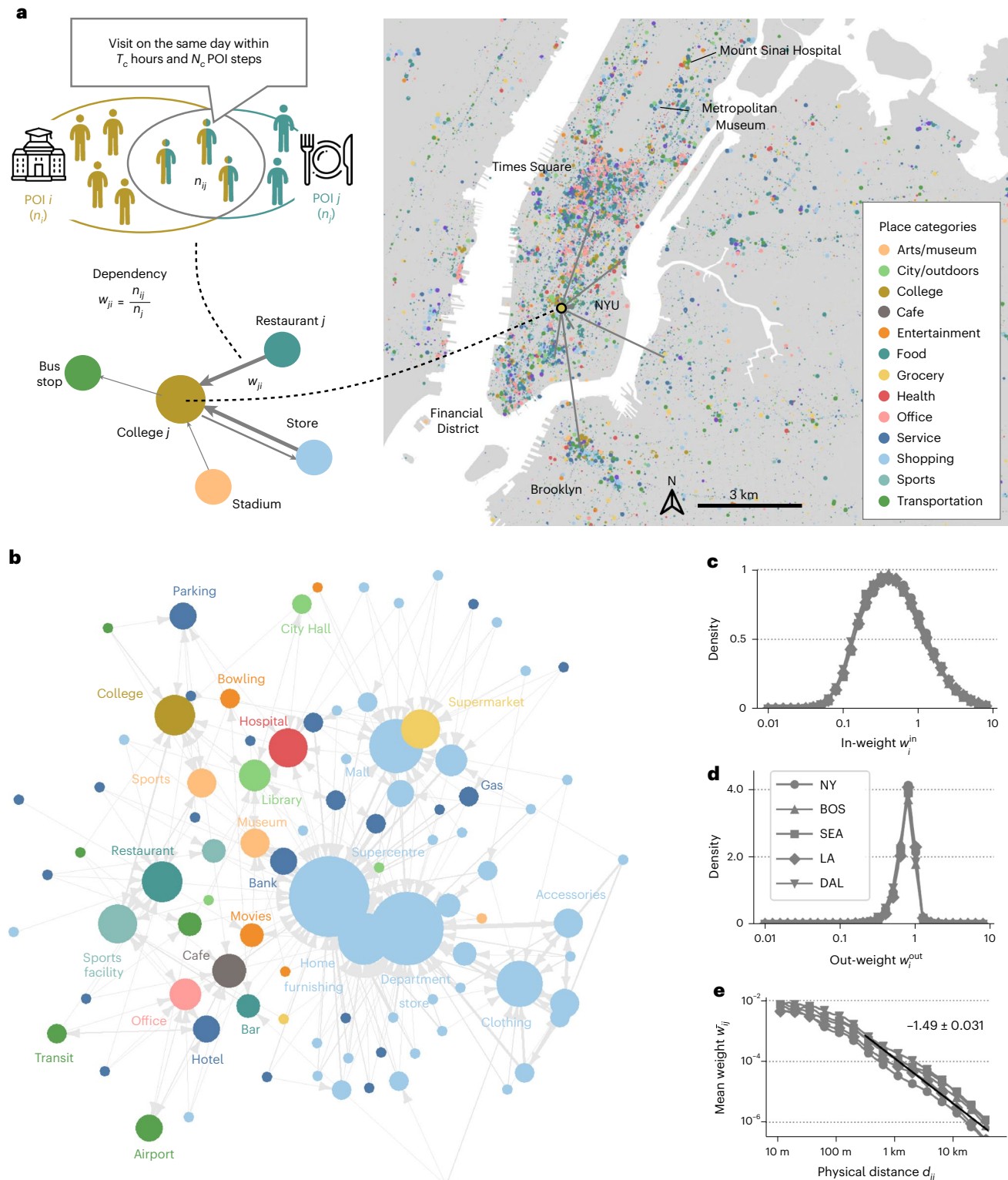

**Fig. 1 | Behaviour-based dependency networks between places in cities.**
**a**, The dependency $w_{ji}$ of POI $j$ (for example, a restaurant) on POI $i$ (for example, a college) is computed as the proportion of the intersection of individuals that visit both the college $i$ and cafe $j$ ($n_{ij}$) on the same day within 6 h ($T_c$) and within 1 step ($N_c$), out of the total count of individuals who visit cafe $j$, $n_j$. Note that dependencies $w_{ij}$ and $w_{ji}$ are asymmetrical and bidirectional. A visualization of total in-weight ($w_i^{in} = \sum_j w_{ji}$) for all POIs in the Manhattan area in New York is shown. The colours show the POI category and the node sizes show the total in-weight. **b**, A network diagram showing the average dependencies between POI subcategories in Boston (other cities shown in Supplementary Note 2). Each node is a POI subcategory, and the three largest outgoing dependency edges are shown

for each node. The node sizes show the in-degree of the constructed network.
**c**,**d**, A probability density distribution of the in-weight $w_i^{in}$ (**c**) and out-weight ($w_i^{out} = \sum_j w_{ij}$) (**d**) per node in the five cities, labeled with different marker shapes. The total in-weight $w_i^{in}$ has a substantially larger variance compared with the out-weight $w_i^{out}$, indicating the existence of nodes with a large attraction of dependencies, such as New York University (NYU, annotated in **a**). **e**, Average dependency $\overline{w_{ij}}$ of all POIs $i$ and $j$ with a Haversine distance of $d_{ij}$. Average dependency decays with $d_{ij}$ with a slope of $\overline{w_{ij}} \propto d_{ij}^{-1.49}$. **a** was designed using icons from Flaticon.com created by Freepik and Education. The maps were produced in Python using the TIGER shapefiles from the US Census Bureau[48].

Figure 1c,d shows the probability density distribution of $w_i^{in}$ and $w_i^{out}$, respectively. Despite the geographical, sociodemographic and economic differences across the five metropolitan areas, we observe striking similarities in the in- and out-weight distributions across cities. The total in-weight $w_i^{in}$ has a substantially larger variance, ranging from 0.01 to 10, compared with the out-weight $w_i^{out}$, which mostly ranges between 0.1 and 1. This is mainly due to the functional form of $w_{ij}$, which allows an arbitrary number of places to depend on a specific place but limits how much one place can depend on others. The long tail of $P(w_i^{in})$ indicates the existence of nodes with a large attraction of dependencies, such as the Metropolitan Museum of Art in New York City (annotated in Fig. 1a), as well as major retail stores such as Walmart, Market Basket and Home Depot (Supplementary Note 2.2 shows a list of places with high $w_i^{in}$ and $w_i^{out}$). The stability of dependency networks are further tested in Supplementary Note 3.1. Supplementary Figs. 18–21 show that the in-weights, out-weights and category pairwise weight proportions are highly correlated (Pearson correlation >0.7) across different time periods (January–April 2019 and May–August 2019) with the baseline time period (September–December 2019).

## Behaviour-based dependency networks are different from colocation networks

'Everything is related to everything else, but near things are more related than distant things', according to Tobler's First Law of Geography[39]. The behaviour-based dependency network is no exception, due to the spatial limits in human mobility[30]. Figure 1e plots the average dependency $\overline{w_{ij}}$ of all POIs $i$ and $j$ separated by a Haversine distance of $d_{ij}$ (in kilometres). In all five metropolitan areas, the average dependency $\overline{w_{ij}}$ of all POIs $i$ and $j$ with a Haversine distance of $d_{ij}$ is relatively constant until around 100 m but decays with $d_{ij}$ in a power-law trend with a slope of $\overline{w_{ij}} \propto d_{ij}^{-1.49}$, estimated by fitting the domain $d_{ij} \in (0.5\,km, 100\,km)$. The slow decay shows that dependency between businesses extends far beyond their local area. However, empirical dependency cannot be described by distance only. To test that idea, we compare empirical dependency networks to null networks generated from principles of physical colocation.

To generate null networks that have similar density and physical characteristics, the total in-weight and in-degree of each node were conserved from the empirical network. The links of the null networks were generated stochastically according to probabilities proportional to the gravity law $g_{ij} = n_i n_j/(d_0 + d_{ij})^\gamma$, where $n_i$ and $n_j$ are the total number of visits to POIs $i$ and $j$, $d_{ij}$ is the physical distance between POIs $i$ and $j$, $d_0$ is the distance cutoff parameter and $\gamma$ is the exponent parameter of the gravity model (Methods). Parameters $d_0$ and $\gamma$ were fitted empirically to maximize the correlation between $g_{ij}$ and $n_{ij}$, which is the total number of common visitors between POIs $i$ and $j$, and were estimated as $d_0 = 0.2$ and $\gamma = 1.5$ (Supplementary Note 4.1). Figure 2a compares the empirical behaviour-based dependency network (left) and the simulated null network (right). Despite controlling for the in-degree and total in-weight of each node, the empirical network is more spatially dispersed compared with the null network, indicating the existence of longer-distance dependencies between places. On the other hand, the null network exhibits clustered local connections around large hubs including university campuses and shopping malls as shown quantitatively using the average clustering coefficients in Fig. 2b, even though all nodes have the same in-degree and in-weight in both networks.

Furthermore, to disentangle the physical and behavioural factors that generate the observed dependency networks, we tested a simple regression model with the specification: $\log w_{ij} \approx \log d_{ij} + n_j + \eta_i + \eta_j + \theta_i + \theta_j$, where $n_j$ is the total number of visits to place $j$, and $\eta_i$ and $\theta_i$ denote the fixed effects of the place category and Public Use Microdata Area (PUMA) of place $i$, respectively. Figure 2c shows the summary of the regression results. The regression results reveal that distance, POI type and neighbourhood, which are all statistically significant ($P < 0.001$) in all cities, only explain around 9–12% of the variance observed in the

dependency weights, suggesting that the empirical behaviour-based dependency network contains much more nuanced and specific information about the relationships between places, which cannot be fully captured using physical factors of the places. Full regression results, as well as robustness tests against the choice of model parameters and selected time period, are shown in Supplementary Note 4 and Supplementary Tables 3–7.

## Predictability of economic resilience via behaviour-based dependency

The results so far show that the behaviour-based dependency networks encode place-specific relationships between businesses and urban amenities that cannot be fully captured by physical characteristics alone. Here, we empirically investigate whether using the dependency network could improve the predictability of how shocks cascade across places in cities, using the COVID-19 pandemic as an example of an extreme empirical shock. The observed change in visits to different places is computed by $\bar{v}_i = (v_i^{COVID}/v_i^{pre} - 1) \times 100\%$, where $v_i^{COVID}$ and $v_i^{pre}$ denote the number of visits to place $i$ during the pandemic (March–June 2020) and the prepandemic period (September–December 2019), respectively. Figure 3a plots the change in visits $\bar{v}_i$ in the Los Angeles metropolitan statistical area, which indicates that most but not all places experienced a negative effect. The probability density distributions for $\bar{v}_i$ for each metropolitan area and for different periods during the pandemic can be found in Supplementary Note 5.1.

Given the dependency between places, we model the change in visits to $i$, $\bar{v}_i$, as the sum of the direct loss of visits that place $i$ experienced due to the pandemic and the network effects from its network neighbours. Network neighbours $j$ are defined as the set of nodes that have a non-zero dependency weight $w_{ij}$. The network effects component, which is the weighted sum of the change in visits that its depended network neighbours $j$ experienced, is described as $\bar{v}_i = \sum_j w_{ij}\bar{v}_j$, as depicted in the schematic in Fig. 3a (right). The dependency weights $w_{ij}$ are computed using mobility covisit data from before the pandemic (September–December 2019). We test whether there is a significant correlation between the change in visits to the ego $\bar{v}_i$ and the weighted sum of the network neighbours $\sum_j w_{ij}\bar{v}_j$ for different place category pairs. Indeed, Fig. 3b shows that most coefficients are significant and positive indicating that loss in visits in the network neighbour nodes is shared with the ego node. Loss of visits to service and shopping stores has the most substantial correlation with other nodes through the behaviour-based dependency network. We further observe that health and office POIs are least affected by the failure of alter nodes, suggesting structural differences in resilience of foot traffic among essential and non-essential POIs.

To demonstrate that these correlations between POI types are not the result of the blanket impact of COVID on all POIs, we consider several alternative null models to predict $\bar{v}_i$ and find that the empirical dependency network is most predictive. Combining the network effects with the fixed effects of the place $i$'s subcategory $\eta_i$ and the PUMA $\theta_i$, we model the change in visits of $i$, $\bar{v}_i$ as

$$\bar{v}_i = \beta_0 + \beta_W \underbrace{\sum_j w_{ij}\bar{v}_j}_{\text{network effects}} + \underbrace{\eta_i + \theta_i}_{\text{ego fixed effects}} + \epsilon, \tag{1}$$

where $\beta_0$ and $\beta_W$ are the estimated coefficients of this regression mode. As a baseline for comparison, we also prepare an alternative model that models the dependency between places based on the gravity-based null network weights $\hat{w}_{ij}$ instead of the empirical dependency weight $w_{ij}$.

Figure 3c shows the adjusted $R^2$ of the normalized visits using (1) area and category fixed effects, (2) fixed effects and gravity-based null model $\hat{w}_{ij}$ and (3) fixed effects with behaviour-based dependency $w_{ij}$. Using the behaviour-based dependency network substantially

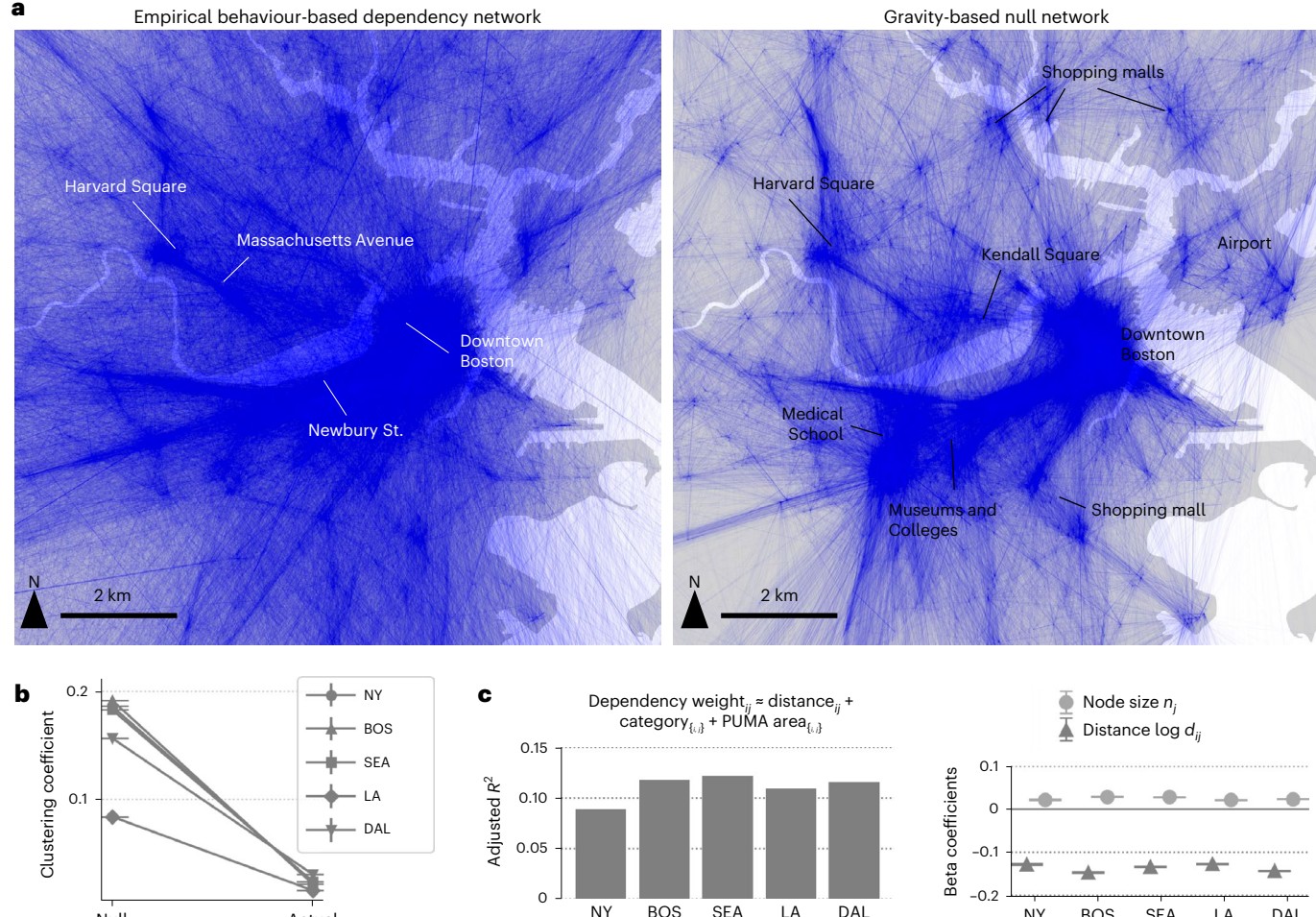

**Fig. 2 | Behaviour-based dependency networks are different from colocation networks. a**, A visual comparison of the empirical behaviour-based dependency network (left) and the simulated null network (right) that stochastically generates network weights between places based on the fitted gravity law, while controlling for the in-degree and the total in-weight of the nodes. Although the number of links and the total weight of each node are consistent, the empirical network is more spatially dispersed compared with the null, indicating the existence of long-distance dependencies between places. On the other hand, the null network exhibits clustered local connections around large hubs, including university campuses and shopping malls. **b**, The null network has a higher average clustering coefficient compared with the empirical network. NY, New York; BOS, Boston; SEA, Seattle; LA, Los Angeles; DAL, Dallas. **c**, Adjusted $R^2$ of the ordinary least squares (OLS) regression model that regresses logged dependency

weight log $w_{ij}$ by physical factors, including the logged distance between POIs $i$ and $j$, the total number of visits and POI subcategories of POIs $i$ and $j$. The low $R^2$ between 0.09 and 0.12 indicates that the dependency weights have distinct characteristics other than physical factors. β coefficients of the size of the node $n_j$ and the physical distance $d_{ij}$ between nodes $i$ and $j$ in the OLS model are statistically significant using a two-sided test ($P < 0.001$). The error bars around the β coefficients indicate the 95% confidence intervals. Dependency weights are larger when the node has more visitors and when the distance is shorter. Full regression results, as well as exact $P$ values and robustness tests against the choice of model parameters and selected time period, are shown in Supplementary Note 4 and Supplementary Tables 3–7. The maps were produced in Python using the TIGER shapefiles from the US Census Bureau[48].

improves the $R^2$ by at least 40% compared with using the physical proximity-based network model, from an average of 0.148 to 0.228. Figure 3d shows the estimated regression coefficients for the physical distance-based dependency ($\hat{\beta}_{null}$) and behaviour-based dependency ($\hat{\beta}_w$) for the five cities and the pooled model. All variables were standardized by subtracting the mean and dividing by the standard deviation for the coefficients to be comparable. Full regression tables are shown in Supplementary Note 5.3. The effects of the behaviour-based dependency are two to three times in magnitude and statistically significant ($P < 0.001$) compared with the physical distance-based dependency, as shown in the regression tables in Supplementary Tables 8–12. Supplementary Note 5.4 shows the estimation results for the different time periods, which all showed better predictability using the behaviour-based dependency network. Additional analysis in Supplementary Note 5.5 showed that using the dependency network further improves the predictability of visitation recovery patterns

(for example, visitation during September–November 2020 compared with March–May 2020). Furthermore, similar results were obtained when using summer breaks as exogenous shocks. The dependency network was able to predict the changes in visits to places that have high dependency on college campuses (Supplementary Note 5.8).

## Cascading impacts of hypothetical urban shocks

Besides the COVID-19 pandemic, what can the behaviour-based dependency network tell us about other types of future shock, such as the increase in online education, remote health services and fewer visits to gas stations due to higher adoption of electrical vehicles? To explore these questions, we apply the network effects model (equation (1)) to simulate the spatial cascades of such shocks in different cities. More specifically, rewriting and reorganizing equation (1) in matrix form, we obtain $\mathbf{v} = W\mathbf{v} + \mathbf{f}$, where $\mathbf{v}$ is a vector of $\bar{v}_i$ for all $N$ places, $W$ is an $N \times N$ matrix where each element is $\tilde{w}_{ij} = \hat{\beta}_W w_{ij}$ and vector $\mathbf{f}$ is an

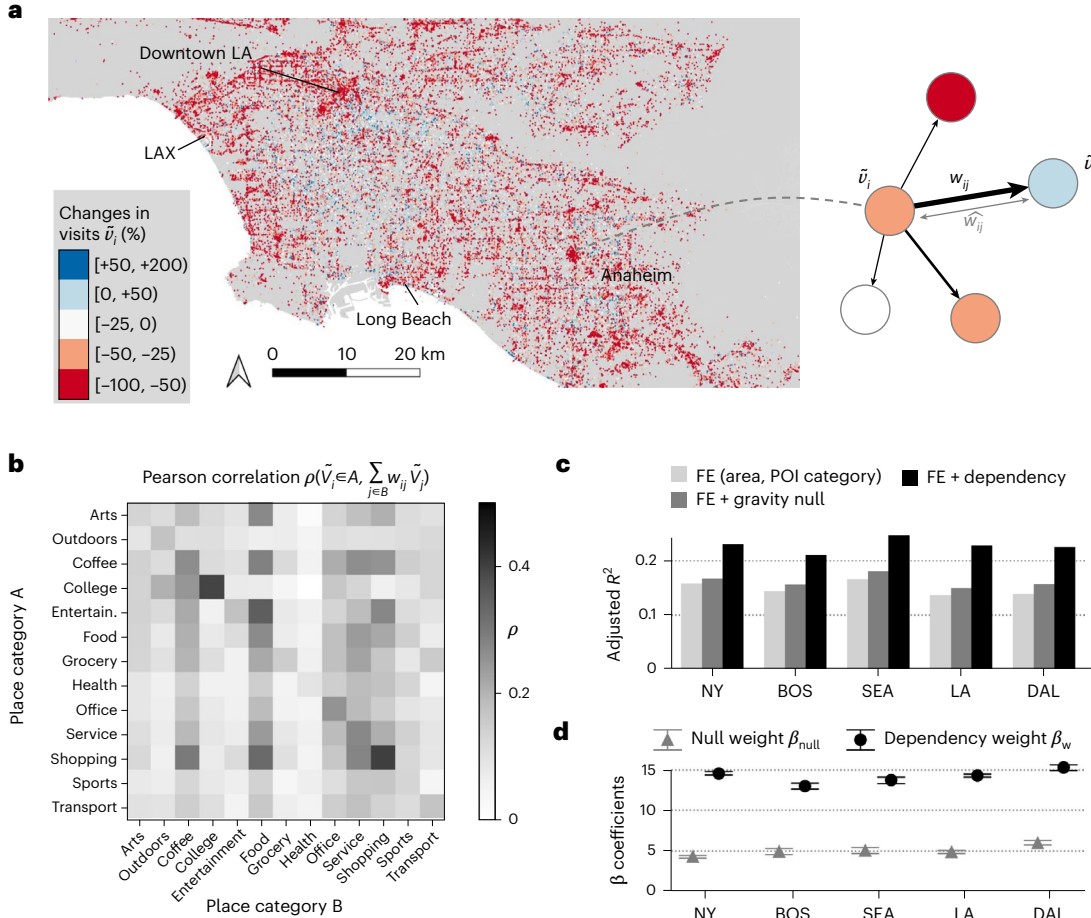

**Fig. 3 | Behaviour-based dependency networks shape economic resilience.**
**a**, A map visualizing the change in visitation patterns to places
$\tilde{v}_i = (v_i^{COVID}/v_i^{pre} - 1) \times 100\%$, during the pandemic (March–May 2020)
compared with the prepandemic period (September–December 2019) in Los
Angeles (LA). LAX, Los Angeles International Airport. Right: the specification of
the model, where the normalized visits at place $i$ are regressed using the sum of
dependency weights weighted by the normalized visits of the network neighbour
nodes, $\sum_j w_{ij}\tilde{v}_j$, and PUMAs and place category fixed effects for POI $i$.
**b**, A correlation between the change in visits to the ego $\tilde{v}_{i\in A}$, which belongs to
category A, and the weighted sum of the network neighbours, which belong to
category B, $\sum_{j\in B} w_{ij}\tilde{v}_j$. Indeed, most coefficients are significant and positive
indicating that loss in visits in the network neighbours is shared with the ego.
**c**, Adjusted $R^2$ of the normalized visits using (1) area and category fixed effects,
(2) fixed effects and gravity model-based dependency weights $\hat{w}_{ij}$ and (3) fixed

effects with behaviour-based dependency weights $w_{ij}$. Using the behaviour-based
dependency network significantly improves the adjusted $R^2$ by 40% (from 0.148
to 0.228) compared with using the distance-based null network. The full results
are shown in the regression tables in Supplementary Tables 8–12. NY, New York;
BOS, Boston; SEA, Seattle; LA, Los Angeles; DAL, Dallas; FE, fixed effects.
**d**, Regression coefficients for the physical distance-based dependency ($\beta_{null}$) and
behaviour-based dependency ($\beta_w$) for the five cities and the pooled model. All
variables were centred and standardized. The error bars show the 95% confidence
interval of the coefficient estimates. The effects of the behaviour-based
dependency are two to three times in magnitude compared with the physical
distance-based dependency and are statistically significant using a two-sided test
at $P < 0.001$. The full results are shown in the regression tables in Supplementary
Tables 8–12. The maps were produced in Python using the TIGER shapefiles from
the US Census Bureau[48].

aggregation of all fixed effects $\beta_0$, $\eta_i$ and $\theta_i$. This model specification is
known as the Leontief open model, which is a simplified and linear
economic model for an economy in which input equals output[40]. To
predict the propagation of shocks throughout places in the city, the
shocks are modelled in the fixed effect vector $f$ (for example, all colleges
experience an external shock where visits are reduced by 50% due to
uptake of online education), and the production vector **v** is computed
by solving the linear system $\hat{\mathbf{v}} = (I - W)^{-1}\mathbf{f}$.

The shift to online education, which occurred during the pan-
demic, is reported to have a continuing effect, with roughly 20% of
school systems planning to or have already started online school pro-
grammes[41]. Previous studies[42], as well as analysis in Fig. 2, have pointed
out that college campuses have a substantial impact on the local econ-
omy. If online learning and remote education were permanent and
increased with the help of advanced technology (for example, aug-
mented reality), what impacts would it have on other businesses and
amenities? Figure 4a shows the simulated effects of a 50% reduction

in visits to college POIs (grey points) on nearby non-college POIs (red
points, the darker red indicates larger negative impacts). Impacted
POIs are limited to those not only in proximity to college POIs but also
in locations that are popular with college students, for example, Mas-
sachusetts Avenue, which connects the Massachusetts Institute of
Technology and Harvard University. For comparison, we simulated the
shocks to non-college POIs using the physical distance network $\hat{W}$,
where $\hat{w}_{ij}$ is used as the matrix elements instead of behaviour-based
dependency $w_{ij}$ (Supplementary Note 6.1). Comparing the simulation
results using the dependency network and the null network shows that
neglecting the behaviour-based dependencies results in a substantial
underestimation of the effects on POIs that are located further away
from colleges.

The effects of online education were heterogeneous for different
place categories located at different distances from colleges. Figure 4b
shows the 90th percentile of impacts on POIs by category and distance
(log scaled). While most substantial impacts occur within 0.5 km, places

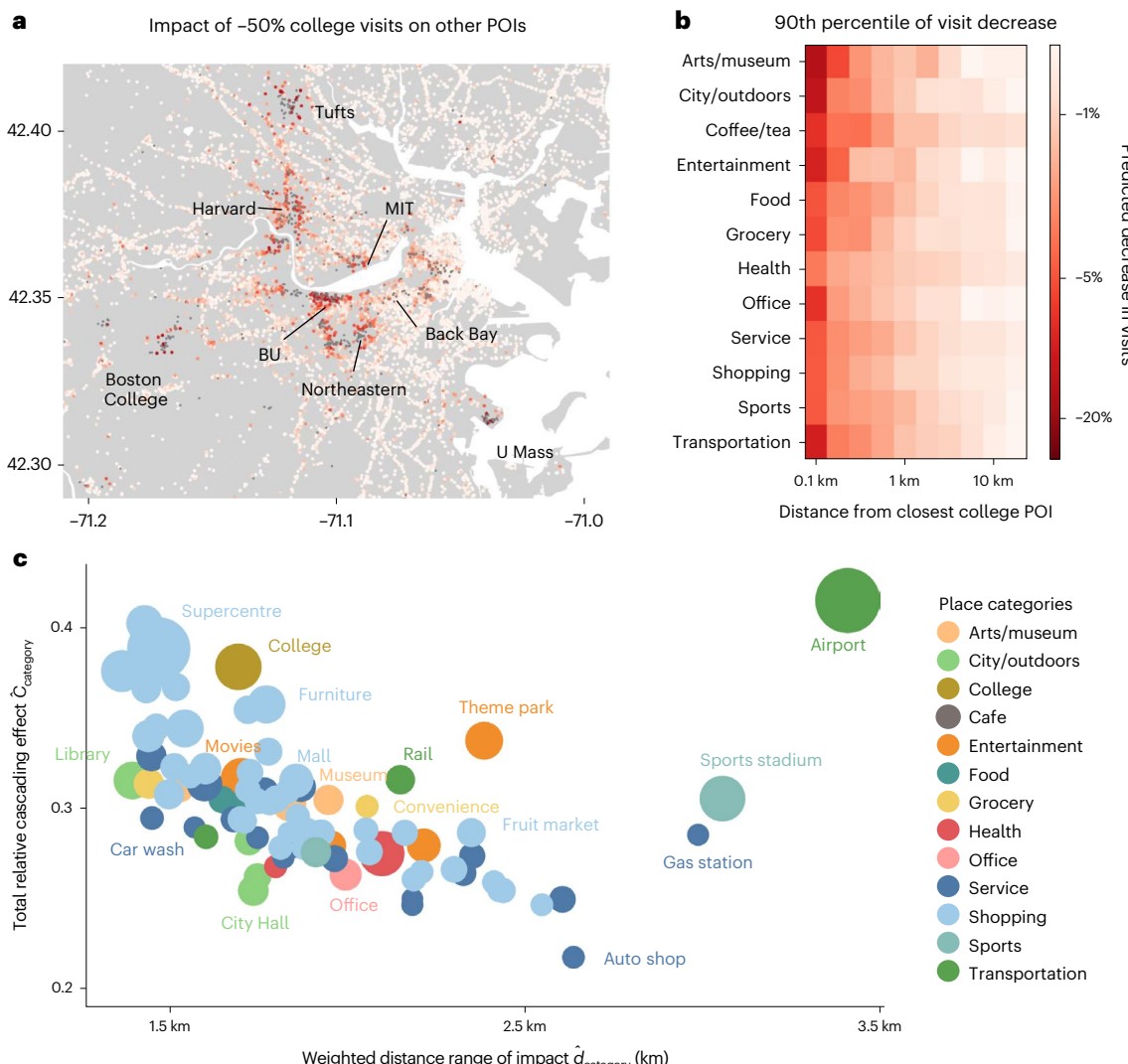

**Fig. 4 | Cascading impacts of hypothetical urban shocks. a**, Simulated effects of a 50% reduction in visits to college POIs (grey points) on nearby non-college POIs (red points, the darker red indicates larger negative impacts), using the fitted Leontief open model. Impacted POIs are limited to those not only in proximity to college POIs but also in locations that are popular with college students. Neglecting the behaviour-based dependencies results in a substantial underestimation of the effects on POIs that are located further away from colleges. MIT, Massachusetts Institute of Technology; BU, Boston University; U Mass, University of Massachusetts. **b**, The impacts of the 50% visit reduction to colleges on places by category and distance (90th percentile decrease in visits are shown, log scaled). While most significant impacts occur within 0.5 km, places such as arts and museums, food and service locations experience substantial long-distance impacts. **c**, Total cascading impact of closing places on other locations, relative to its own size (x axis) and the weighted distance range of the impact (y axis) for different POI subcategories. The node sizes indicate the average number of visitors per POI. Supercentres and colleges have high cascading effects but are focused locally (~1.5 km around the POI). On the other hand, the impacts of airports, stadiums, theme parks and gas stations are both large and far reaching (around 2.5–3.5 km). The maps were produced in Python using the TIGER shapefiles from the US Census Bureau[48].

---

such as arts and museums, food and service places experience substantial long-distance impacts. Simulations assuming different levels of visit decrease to colleges (for example, −100%, −25%) show a similar long-distance cascade of shocks (Supplementary Note 6.1). These persistent spatial cascades emphasize the importance of considering behaviour-based dependency relationships between places to grasp the holistic impact of such urban shocks for resilient urban planning.

Further leveraging the network model, we are able to simulate the impacts of POI closure scenarios and identify the seed nodes (POIs) that have the largest cascading effects on other POIs if inflicted by other urban shocks. For each node, we simulate the cascading impacts of a 100% visit change to a single POI $i$, by computing $\hat{\mathbf{v}}^{(i)} = (I - W)^{-1}\mathbf{e}^{(i)}$, where $\mathbf{e}^{(i)}$ is a one-hot encoding vector of the initial shock that assigns a change in visits of +1 to node $i$ and 0 otherwise, and $\hat{\mathbf{v}}^{(i)}$ is the resulting vector of the cascading impacts, where each element measures the impacts of the initial shock to all nodes. The total impacts of changes

in the number of visits to all nodes can be computed by multiplying $\hat{\mathbf{v}}^{(i)} = (\hat{v}_1^{(i)}, \dots, \hat{v}_N^{(i)})$ with the vector of total visits to each POI, $\mathbf{n} = (n_1, \dots, n_N)$. Thus, the total impacts of the initial shock to node $i$ can be computed by $C_i = \sum_{j;j \neq i} \hat{v}_j^{(i)} n_j$. By further scaling the impact to its own size $n_i$, we obtain the total relative cascading effect as $\hat{C}_i = C_i/n_i$. For example, $\hat{C}_i = 0.3$ would indicate that increasing the number of visits to node $i$ by 100% (or equal to $n_i$) results in a total of $30\% \times n_i$ increase in visits across all other nodes. The mean relative cascading impacts of each POI category, $\hat{C}_{\text{category}}$ are shown in the $y$ axis of Fig. 4c. POI categories such as airports, supercentres, colleges, furniture stores, theme parks, railway stations and sports stadiums have a high impact on other POIs in urban areas propagated through behaviour-based dependency networks.

When implementing policies to close down certain POIs for emergency response (for example, lockdowns during pandemics), it is important to understand the spatial extent of the cascade. To quantify

this, we defined the distance range of the cascade by computing the average distance to impacted nodes, weighted by the magnitude of the impacts. More specifically, we compute the weighted distance range of POI $i$ by $\hat{d}_i = \sum_{j;j\neq i} \hat{v}_j^{(i)} d_{ij} / \sum_{j;j\neq i} \hat{v}_j^{(i)}$. The weighted distance range of impact for each POI category, $\hat{d}_{\text{category}}$ are shown in the $x$ axis of Fig. 4c. Supercentres and colleges have high cascading effects but are focused locally (-1.5 km around the POI). On the other hand, the impacts of airports, stadiums, theme parks and gas stations are both large and far reaching (around 2.5–3.5 km). Estimation results for all cities are shown in Supplementary Note 6.2. Understanding the magnitude and spatial extent of the cascading effects could be applied to design emergency management policies to effectively close places while minimizing economic losses. The large magnitude of the spatial cascades that occur due to behaviour-based dependency networks calls for new urban policy-making approaches that balance the benefits of mobility restriction measures (for example, preventing the spread of diseases) while minimizing the total cascading economic impacts to urban places and amenities.

## Discussion

Fostering the resilience of urban systems to shocks is an urgent challenge for cities and communities, with increasing risks of climate change-induced disasters, long-lasting effects of the COVID-19 pandemic and unprecedented technological shifts in how we move (electric vehicles and autonomous vehicles), work[43], shop and learn[44]. Such urban shocks could induce substantial shifts in human behaviour and urban activity patterns by changing the various incentive and cost mechanisms that motivate activities in cities. A plethora of research has focused on modelling the spillover effects of disruptions on supply chains across industries (for example, ref. 5); however, there has been limited investigation into the spillover effects that could be mediated through the movement of people in cities. The broader socioeconomic impacts that such behavioural changes could have on cities, for example, on the social fabric of communities[7], economic networks and local businesses and urban infrastructure systems, are not well understood. Motivated by these critical challenges, we used empirical data from mobile phone devices collected from five major metropolitan areas in the United States to quantitatively measure and analyse the urban economic networks mediated by human behaviour and their resilience to potential urban shocks.

In this context, our study contains three important contributions towards understanding the economic network dependencies in cities. First, our approach measures and reveals that the dependent relationships that exist between businesses and places are highly complex products of human behavioural preferences and decisions rather than a measure determined solely by the urban form, including the physical distance between places, and their physical locations, popularity and categories (which only explain around 10% of the variance). We observe the existence of places that are highly dependent on hundreds of other places (for example, gas stations and gyms) and others that are depended by many other locations (for example, major shopping centres, universities and major hospitals). Our results show that urban economic networks are determined by the behaviour generated by individual activity patterns, connecting distant businesses and amenities due to the combination of work, leisure and shopping activities on the same day. However, businesses that are next door to each other are not necessarily dependent on each other, since they target and attract people with different interests and lifestyles. Second, using different periods of the COVID-19 pandemic as external shocks, we showed that using the prepandemic behaviour-based dependency network with a Leontief input–output formulation improves the predictability of the spillover shocks to different businesses, compared with the distance-based colocation network. Third, simulations of hypothetical urban shocks showed that the dependencies generated by human

behaviour significantly amplify the shocks to places that are located further away from the origins of the shocks. Our results show that, for example, while supercentres affect mostly local POIs, airports, sports stadiums and gas stations have a substantial long-range effect on POIs across the city. Policies to contain the spread of pandemics or future urban shocks need to incorporate the spatial extent of the impacts mediated by the dependency network. This points to the importance of shifting from a place-based approach to a network-based approach in designing urban interventions and, in general, in understanding the socioeconomic impacts of urban changes.

Our study has several limitations. First, dependency weights between places were computed using all mobile phone users that were observed to visit both places ('covisit'); however, we were not able to differentiate between visitations of different natures. We could further classify covisits into different types, such as routine and exploration behaviour. Moreover, following recent studies focusing on the substantial differences in mobility behaviour across sociodemographic groups (for example, refs. 45,46), we could decompose the dependencies into different income ranges to better understand which sociodemographic segments are contributing to different types of dependency relationship. Therefore, the dependency metric computed in our study should be interpreted as an aggregated measure of all covisits that occur in cities, and further decomposition and contextualization of the dependency metric could be conducted when applying this approach to analyse specific urban shocks and policies. Second, another challenge lies in understanding how the dependency networks between places reorganize due to various urban shocks. The assumption in this study did not consider such dynamic reorganization because of the sudden and short-term nature of the shocks we analysed (for example, COVID-19 and closures of colleges). Modelling the dynamics of the behaviour-based dependency networks using human behaviour data observed across a longer time frame, and applying the method to a broader range of realistic urban disruptions including climate change-induced disasters could be an interesting topic for future research. Third, in this study, simple linear models (for example, the Leontief open model) were used to test the effectiveness of our approach to modelling economic resilience. Given the advancements in nonlinear and complex models for graph structured data (for example, graph neural networks[47]), there is potential for future works that develop models that improve the predictability of resilience and are capable of describing the microscopic temporal dynamics of disruption and recovery. Four, this method measures the dependency relationships between places through foot traffic patterns, and this may not capture the full breadth of economic interactions. This approach simplifies complex economic relationships and may not capture other channels of economic interactions, such as online transactions or services not tied to physical visits. Follow-up work using other behaviour datasets, such as credit card purchase data, would be an interesting avenue for future research[37].

Our findings have implications for our understanding of the resilience of urban systems to shocks. While spillover effects of urban shocks (for example, disasters and power outages) have been often studied with a focus on supply chains that connect firms and industries, our results show that human behaviour and mobility patterns also contribute to the cascade of shocks across businesses and amenities in different industries. This study shows robust results on the predictability of economic resilience, laying the groundwork for future investigations into causal effects of dependencies on economic resilience (for example, through natural experiments around natural disasters, rainfall or construction projects that impact mobility). To better understand how the effects of urban shocks or technological shifts would manifest in cities, a spatial understanding of risk (which in itself underestimates the broader impacts of shocks) should be complemented by how the flow of individuals connects different firms and places. This framework could be applied to assess the impacts of both negative and positive

shocks on cities. Examples of negative shocks include natural hazards and pandemics, while urban transportation (for example, fare-free bus programmes) and land use policies (for example, pedestrian-only streets) could have positive shocks on businesses in the neighbourhoods. Urban planners could leverage the observed dependency relationships to target such policies, mitigate potential disruptions better and amplify the positive impacts on local businesses and the wellbeing of communities. It also calls for a more holistic understanding of shopping, innovation or shopping districts, since the vibrancy of those places and their impact on other areas might depend on business and amenities across the city.

## Methods

### Mobility and POI data

We utilize an anonymized location dataset of mobile phones and smartphone devices provided by Spectus, a location data intelligence company that collects anonymous, privacy-compliant location data of mobile devices using their software development kit technology in mobile applications and privacy framework. Spectus processes data collected from mobile devices whose owners have actively opted in to share their location and require all application partners to disclose their relationship with Spectus, directly or by category, in the privacy policy. With this commitment to privacy, the dataset contains location data for roughly 15 million daily active users in the United States. All data analysed in this study are aggregated to preserve privacy. To measure the visitation patterns of individuals in urban environments, we attribute the stops of individual users to specific places in the city. To study the stops at different places, we use stops that are longer than 10 min but shorter than 10 h. In our study, we use location data of places collected by Safegraph. To protect the users' privacy, we have removed various privacy-sensitive places from our places database, including health-related places, places where the vulnerable population are located, military-related, religious facilities, places that are related to sexual orientation and adult-oriented places. As a result, we have a total of over 1 million places in the five cities. The home locations of individual users are estimated at the Census Block Group level using different variables including the number of days spent in a given location in the last month, the daily average number of hours spent in that location and the time of the day spent in the location during nighttime (see Supplementary Note 1.1 for more details). The representativeness of this data has been tested and corrected in Supplementary Notes 1.3 and 1.4 using poststratification techniques. Since the data used were anonymized and spatially aggregated at places, categories or census areas, we were granted an exemption by the Massachusetts Institute of Technology Committee on the Use of Humans as Experimental Subjects (COUHES protocol no. 1812635935) and its extension no. E-2962.

### Behaviour-based dependency networks

We define the dependence of a POI $i$ on another POI $j$ as $w_{ij} = \frac{n_{ij}}{n_i}$, where $n_i$ denotes the number of visits to POI $i$ and $n_{ij}$ denotes the number of 'covisits' between POIs $i$ and $j$. A covisit is defined as an instance in which POIs $i$ and $j$ were visited by the same individual (1) on the same day, (2) within $T_c$ (threshold parameter, $T_c = 6$ hours used in main results) hours from exiting POI $i$ ($j$) to entering POI $j$ ($i$) and (3) within $N_c$ intermediate POIs (threshold parameter; $N_c = 1$ is used in main results). Because the denominator is based on the number of visits to the target POI, $w_{ij} \neq w_{ji}$. This simple but intuitive measure considers the asymmetric nature of dependencies between POIs. By computing the dependency weights $w_{ij} \forall i, j$, we obtain the behaviour-based dependency matrix $W \in \mathbb{R}^{N \times N}$, where $N$ is the total number of POIs present in the CBSA. As a baseline parameter setting, we use $T_c = 6$ h and $T_s = 1$ POI. The sensitivity and statistical robustness of the dependency network when using different covisit detection threshold parameters $T_c$ and $T_s$, only short non-work visits and data from different time periods are tested and discussed in Supplementary Notes 2 and 3.

### Distance-based null networks

To generate null networks that preserve basic structural properties, (1) the weight $w_{ij}$ decays with physical distance and (2) the in-weight $w_{ij}$ is larger for nodes with larger visitation $n_i$, we generated edges based on the generalized gravity law: $g_{ij} = n_i n_j / (d_0 + d_{ij})^\gamma$, where $n_i$ and $n_j$ are the total number of visits to POIs $i$ and $j$, $d_{ij}$ is the physical distance between POIs $i$ and $j$, $d_0$ is the distance cutoff parameter and $\gamma$ is the exponent parameter of the gravity model. Parameters $d_0 = 0.2$ and $\gamma = 1.5$ were fitted empirically to maximize the correlation between $g_{ij}$ and $n_{ij}$, which is the total number of common visitors between POIs $i$ and $j$ (Supplementary Note 4.1). For each edge in the actual dependency network connecting $i$ and $j$ with a dependency weight $w_{ij}$, we compute its gravity component using the empirical fit between $g_{ij}$ and $w_{ij}$ and select an alternative node with the same level of corresponding gravity weight from its 10,000 closest nodes and is assigned the same weight $w_{ij}$. This algorithm enables us to construct a null network where we (1) maintain the linear relationship between $w_{ij}$ and $g_{ij}$, (2) the same number of in-edges are selected for each node and (3) the total in-weight for each node is kept consistent. The details about the null network generation procedure, as well as statistical analysis of the null network and their differences from the actual dependency network, can be found in Supplementary Note 4.

### Modelling impacts of COVID-19 using dependency networks

To investigate the utility of the dependency network for predicting the resilience of businesses, we construct regression models that predict the change in visitation patterns to a POI using information about the change in visitation patterns to its alters and the dependency network. The observed change in visits to different places is computed by $\bar{v}_i = v_i^{\text{after}} / v_i^{\text{before}} - 1 \times 100(\%)$, where $v_i^{\text{before}}$ and $v_i^{\text{after}}$ denote the number of visits to place $i$ before the pandemic (September–December 2019) and during different periods of the pandemic period (March–November 2020), respectively. We build a simple linear regression model of the form:

$$\bar{v}_i \approx \sum_j w_{ij} \bar{v}_j + \sum_j \hat{w}_{ij} \bar{v}_j + \eta_i + \theta_i, \tag{2}$$

where $\bar{v}_i$ denotes the change in visitations to POI $i$ during the different stages of the pandemic in 2020, $\sum_j w_{ij} \bar{v}_j$ is the sum of the network neighbours' (POIs $j$) change in visitations ($\bar{v}_j$) weighted by the dependency network weights, $w_{ij}$, $\sum_j \hat{w}_{ij} \bar{v}_j$ is the sum of the network neighbours' (POIs $j$) change in visitations ($\bar{v}_j$) weighted by the distance-based null network weights, $\hat{w}_{ij}$, $\eta_i$ is the fixed effect for POI $i$'s subcategory, and $\theta_i$ is the fixed effect for POI $i$'s located PUMA. Robustness of regression results to the choice of the time period used to generate the dependency network, different model parameters and similar results obtained using the case study of college summer breaks are shown in Supplementary Note 5.

### Simulating cascades of hypothetical urban shocks

To simulate the spatial cascades of shocks in different cities, we use the model specification of the Leontief open model. Rewriting and reorganizing the regression model in matrix form, we obtain $\mathbf{v} = W\mathbf{v} + \mathbf{f}$, where $\mathbf{v}$ is a vector of $\bar{v}_i$ for all $N$ places, $W$ is an $N \times N$ matrix, where each element is $\tilde{w}_{ij} = \hat{\beta}_W w_{ij}$, and vector $\mathbf{f}$ is an aggregation of all fixed effects $\beta_0$, $\eta_i$ and $\theta_i$. To predict the propagation of shocks throughout places in the city, the shocks are modelled in the fixed effect vector $f$ (for example, all colleges experience an external shock of −50% visits reduction due to uptake of online education), and the production vector $\mathbf{v}$ is computed by solving the linear system $\hat{\mathbf{v}} = (I - W)^{-1}\mathbf{f}$ via the generalized minimal residual iteration method. Details and parameter sensitivity analysis can be found in Supplementary Note 6.1.

Furthermore, we simulate the impacts of POI closure scenarios and identify the seed nodes (POIs) that have the largest cascading

effects on other POIs if inflicted by other urban shocks. For each node, we simulate the cascading impacts of a 100% visit change to a single node $i$, by computing $\hat{\mathbf{v}}^{(i)} = (I - W)^{-1}\mathbf{e}^{(i)}$, where $\mathbf{e}^{(i)}$ is a one-hot encoding vector of the initial shock that assigns a change in visits of +1 to node $i$ and 0 otherwise, and $\hat{\mathbf{v}}^{(i)}$ is the resulting vector of the cascading impacts, where each element measures the impacts of the initial shock to all nodes. The total impacts of changes in the number of visits to all nodes can be computed by multiplying $\hat{\mathbf{v}}^{(i)} = (\hat{v}_1^{(i)}, \dots, \hat{v}_N^{(i)})$ with the vector of total visits to each POI, $\mathbf{n} = (n_1, \cdots, n_N)$. Thus, the total impacts of the initial shock to node $i$ can be computed by $C_i = \sum_{j; j \neq i} \hat{v}_j^{(i)} n_j$. By further scaling the impact to its own size $n_i$, we obtain the total relative cascading effect as $\hat{C}_i = C_i / n_i$. The distance range of the cascade is computed as the average distance to impacted nodes, weighted by the magnitude of the impacts. More specifically, we compute the weighted distance range of POI $i$ by $\hat{d}_i = \sum_{j; j \neq i} \hat{v}_j^{(i)} d_{ij} / \sum_{j; j \neq i} \hat{v}_j^{(i)}$. Figure 4c plots the relative cascading impacts and distance ranges of each POI category, $\hat{C}_{\mathrm{category}}$ and $\hat{d}_{\mathrm{category}}$. More details and results for all cities can be found in Supplementary Note 6.2.

### Reporting summary

Further information on research design is available in the Nature Portfolio Reporting Summary linked to this article.

## Data availability

The data that support the findings of this study are available from Spectus through their Social Impact programme, but restrictions apply to the availability of these data, which were used under the license for the current study and are, therefore, not publicly available. Information about how to request access to the data and its conditions and limitations can be found in https://spectus.ai/social-impact/. Data access requests should be submitted through Spectus' Social Impact customer page via https://spectus.ai/lp/book-a-demo/, where the sales team at Spectus may be contacted in a timely manner. Data about the POI locations were provided by Safegraph, who can be contacted through https://www.safegraph.com/. The Safegraph data are available through the Dewey platform through a paid subscription via https://app.deweydata.io/home. Tiger shapefiles can be downloaded from the US Census Bureau via https://www.census.gov/programs-surveys/geography/guidance/tiger-data-products-guide.html.

## Code availability

The analysis was conducted using Python. The code to reproduce the main results in the figures from the aggregated data is publicly available on GitHub via https://github.com/takayabe0505/dependencynetwork.

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

## Acknowledgements

We thank Spectus, who kindly provided us with the mobility data set for this research through their Data for Good programme. T.Y. acknowledges support by the National Science Foundation under grant number 2425021. E.M. acknowledges support by Ministerio de Ciencia e Innovación/Agencia Española de Investigación (MCIN/AEI/10.13039/501100011033) through grant PID2019-106811GB-C32 and the National Science Foundation under grants 2218748 and 2420945. The funders had no role in study design, data collection and analysis, decision to publish or preparation of the manuscript.

## Author contributions

T.Y. designed the algorithms, performed the analysis, and developed models and simulations. B.G.B.B. and E.M. performed part of the analysis and partially developed models and simulations. M.F., A.P. and E.M. supervised the research. All authors wrote the paper. The company data were processed by T.Y., B.G.B.B. and E.M. All authors had access to aggregated (non-individual) processed data. All authors reviewed the manuscript.

## Competing interests

The authors declare no competing interests.

## Additional information

**Correspondence and requests for materials** should be addressed to Takahiro Yabe or Esteban Moro.

Esteban Moro

# Reporting Summary

## Statistics

For all statistical analyses, confirm that the following items are present in the figure legend, table legend, main text, or Methods section.

| n/a | Confirmed | |
|---|---|---|
| ☐ | ☒ | The exact sample size ($n$) for each experimental group/condition, given as a discrete number and unit of measurement |
| ☒ | ☐ | A statement on whether measurements were taken from distinct samples or whether the same sample was measured repeatedly |
| ☐ | ☒ | The statistical test(s) used AND whether they are one- or two-sided<br>*Only common tests should be described solely by name; describe more complex techniques in the Methods section.* |
| ☐ | ☒ | A description of all covariates tested |
| ☒ | ☐ | A description of any assumptions or corrections, such as tests of normality and adjustment for multiple comparisons |
| ☐ | ☒ | A full description of the statistical parameters including central tendency (e.g. means) or other basic estimates (e.g. regression coefficient) AND variation (e.g. standard deviation) or associated estimates of uncertainty (e.g. confidence intervals) |
| ☐ | ☒ | For null hypothesis testing, the test statistic (e.g. $F$, $t$, $r$) with confidence intervals, effect sizes, degrees of freedom and $P$ value noted<br>*Give P values as exact values whenever suitable.* |
| ☒ | ☐ | For Bayesian analysis, information on the choice of priors and Markov chain Monte Carlo settings |
| ☒ | ☐ | For hierarchical and complex designs, identification of the appropriate level for tests and full reporting of outcomes |
| ☐ | ☒ | Estimates of effect sizes (e.g. Cohen's $d$, Pearson's $r$), indicating how they were calculated |

*Our web collection on statistics for biologists contains articles on many of the points above.*

## Software and code

Policy information about availability of computer code

| Data collection | No special software was used to collect the data. |
|---|---|
| Data analysis | Data analysis was conducted using different python libraries. Here is a list of references, included in the Supplementary Material.<br>- Charles R Harris, K Jarrod Millman, Stefan J Van Der Walt, Ralf Gommers, Pauli Virtanen, David Cournapeau, Eric Wieser, Julian Taylor, Sebastian Berg, Nathaniel J Smith, et al. Array programming with numpy. Nature, 585(7825):357–362, 2020. (version 1.24.2)<br>- John D Hunter. Matplotlib: A 2d graphics environment. Computing in Science & Engineering, 9(03):90–95, 2007. (version 3.7.0)<br>- Wes McKinney et al. pandas: a foundational python library for data analysis and statistics. Python for high performance and scientific computing, 14(9):1–9, 2011. (version 1.5.3)<br>- K Jordahl. Geopandas: Python tools for geographic data. URL: https://github.com/geopandas/geopandas, 3, 2014.(version 0.12.2)<br>- Skipper Seabold and Josef Perktold. Statsmodels: Econometric and statistical modeling with python. In Proceedings of the 9th Python in Science Conference, volume 57, page 61. Austin,TX, 2010. (version 0.13.5)<br>- Stargazer implementation in R. URL: https://github.com/mwburke/stargazer (Python implementation of the R stargazer multiple regression model creation tool, version 5.2.3)<br><br>Code to reproduce our results in the figures from the aggregated data will be available on Github. URL: https://github.com/takayabe0505/dependencynetwork |

For manuscripts utilizing custom algorithms or software that are central to the research but not yet described in published literature, software must be made available to editors and reviewers. We strongly encourage code deposition in a community repository (e.g. GitHub). See the Nature Portfolio guidelines for submitting code & software for further information.

# Data

Policy information about availability of data

All manuscripts must include a data availability statement. This statement should provide the following information, where applicable:
- Accession codes, unique identifiers, or web links for publicly available datasets
- A description of any restrictions on data availability
- For clinical datasets or third party data, please ensure that the statement adheres to our policy

The data that support the findings of this study are available from Spectus through their Social Impact program, but restrictions apply to the availability of these data, which were used under the license for the current study and are therefore not publicly available. Information about how to request access to the data and its conditions and limitations can be found in https://spectus.ai/social-impact/ . Data access requests should be submitted through Spectus' Social Impact customer page https://spectus.ai/lp/book-a-demo/. Timely response should be expected from the Sales team at Spectus. Data about the points-of-interest locations was provided by Safegraph, who can be contacted through https://www.safegraph.com/. The Safegraph data is available through the Dewey platform through a paid subscription https://app.deweydata.io/home. Tiger shapefiles can be downloaded from the US Census Bureau https://www.census.gov/programs-surveys/geography/guidance/tiger-data-products-guide.html.

# Research involving human participants, their data, or biological material

Policy information about studies with human participants or human data. See also policy information about sex, gender (identity/presentation), and sexual orientation and race, ethnicity and racism.

| | |
|---|---|
| Reporting on sex and gender | The data contains no information about sex or gender. |
| Reporting on race, ethnicity, or other socially relevant groupings | The data or results contains no information about race or ethnicity. |
| Population characteristics | Data used are geo-locations from anonymous opted-in devices collected by the company Spectus in four metro areas in the US. Data have been aggregated at the level of places, categories, or census areas where a number of devices are present to prevent de-anonymization. The data is quantitative as it reflects the precise time and geolocation of the anonymous users or metrics from the census. |
| Recruitment | Data used are geo-locations from anonymous opted-in devices collected by the company Spectus in four metro areas in the US. Data have been aggregated at the level of places, categories, or census areas where a number of devices are present to prevent de-anonymization. Visitation information at sensitive places have been removed to protect the privacy of individuals. To minimize the potential bias in the geographical penetration of the users, we have implemented post-stratification techniques. All details about our sampling methods and post-stratification methods can be found in the Supplementary Material. |
| Ethics oversight | Since the data used was anonymized and spatially aggregated at places, categories, or census areas, we were granted an Exemption by the MIT Committee on the Use of Humans as Experimental Subjects (COUHES protocol #1812635935) and its extension #E-2962. |

Note that full information on the approval of the study protocol must also be provided in the manuscript.

# Field-specific reporting

Please select the one below that is the best fit for your research. If you are not sure, read the appropriate sections before making your selection.

☐ Life sciences  ☒ Behavioural & social sciences  ☐ Ecological, evolutionary & environmental sciences

For a reference copy of the document with all sections, see nature.com/documents/nr-reporting-summary-flat.pdf

# Behavioural & social sciences study design

All studies must disclose on these points even when the disclosure is negative.

| | |
|---|---|
| Study description | The study analyzes geo-location data from anonymous opted-in devices collected by the company Spectus in 5 metro areas in the US to understand the visitation patterns to different places, businesses, and amenities. Data have been aggregated at the level of places, categories, or census areas where a number of devices are present to prevent de-anonymization. The data enabled analysis of urban dynamics via quantitative methods. |
| Research sample | The sample of users is described above: anonymous opted-in devices collected by the company Spectus. The study sample was chosen to understand the microscopic behavior of individuals in urban environments at a high spatial and temporal granularity. The four cities were chosen to observe differences between cities with large variability in sociodemographic, geographic, policy, climate, and urban form. To maximize the representativeness of the data, we implemented post-stratification techniques. The details of the methods can be found in the Supplementary Material. |

| Sampling strategy | Mobile phone users whose mobility activity was sufficiently observed in the dataset during the period of interest. See the Methods section and Supplementary Material for the details of the sampling strategy. The sample size and percentage calculation was performed by overlaying the mobile phone location data with census data from the American Community Survey. The sample is sufficient since we performed post-stratification to minimize potential biases in the data sample and the geographical differences in mobile phone penetration rate of users. Details may be found in the Supplementary Material. |
|---|---|
| Data collection | Data was collected by the company Spectus through different applications on their mobile phones. The researchers were blinded to experimental conditions and the study hypothesis. |
| Timing | Data was collected through January 2019 to January 2022. |
| Data exclusions | No data was excluded |
| Non-participation | Only anonymous opted-in devices were used in the analysis. |
| Randomization | The data collected is observational and does not come from an experiment. Thus, this is not applicable. |

# Reporting for specific materials, systems and methods

We require information from authors about some types of materials, experimental systems and methods used in many studies. Here, indicate whether each material, system or method listed is relevant to your study. If you are not sure if a list item applies to your research, read the appropriate section before selecting a response.

## Materials & experimental systems

| n/a | Involved in the study |
|---|---|
| ☒ | Antibodies |
| ☒ | Eukaryotic cell lines |
| ☒ | Palaeontology and archaeology |
| ☒ | Animals and other organisms |
| ☒ | Clinical data |
| ☒ | Dual use research of concern |
| ☒ | Plants |

## Methods

| n/a | Involved in the study |
|---|---|
| ☒ | ChIP-seq |
| ☒ | Flow cytometry |
| ☒ | MRI-based neuroimaging |

## Plants

| Seed stocks | *Report on the source of all seed stocks or other plant material used. If applicable, state the seed stock centre and catalogue number. If plant specimens were collected from the field, describe the collection location, date and sampling procedures.* |
|---|---|
| Novel plant genotypes | *Describe the methods by which all novel plant genotypes were produced. This includes those generated by transgenic approaches, gene editing, chemical/radiation-based mutagenesis and hybridization. For transgenic lines, describe the transformation method, the number of independent lines analyzed and the generation upon which experiments were performed. For gene-edited lines, describe the editor used, the endogenous sequence targeted for editing, the targeting guide RNA sequence (if applicable) and how the editor was applied.* |
| Authentication | *Describe any authentication procedures for each seed stock used or novel genotype generated. Describe any experiments used to assess the effect of a mutation and, where applicable, how potential secondary effects (e.g. second site T-DNA insertions, mosiacism, off-target gene editing) were examined.* |

