## [Peer Review File · Nature Human Behaviour]

Behavior-based dependency networks between places shape urban economic resilience

Corresponding Author: Dr Takahiro Yabe

Version 0:

Decision Letter:

6th March 2024

Dear Taka,

Thank you once again for your manuscript, entitled "Behavior-based dependency networks between places shape urban economic resilience", and for your patience during the peer review process.

Your Article has now been evaluated by 3 referees. You will see from their comments copied below that, although they find your work of potential interest, they have raised quite substantial concerns. In light of these comments, we cannot accept the manuscript for publication, but would be interested in considering a revised version if you are willing and able to fully address reviewer and editorial concerns.

We hope you will find the referees' comments useful as you decide how to proceed. If you wish to submit a substantially revised manuscript, please bear in mind that we will be reluctant to approach the referees again in the absence of major revisions. We are committed to providing a fair and constructive peer-review process. Do not hesitate to contact us if there are specific requests from the reviewers that you believe are technically impossible or unlikely to yield a meaningful outcome.

To guide the scope of the revisions, the editors discuss the referee reports in detail within the team, including with the chief editor, with a view to (1) identifying key priorities that should be addressed in revision and (2) overruling referee requests that are deemed beyond the scope of the current study. We hope that you will find the prioritised set of referee points to be useful when revising your study. Please do not hesitate to get in touch if you would like to discuss these issues further.

In particular, please address the following (as well as all other reviewer comments):

- 1) You will see from their comments that Reviewer 1 calls for a number of additional analyses, as well as extension of your investigation to other disaster settings. Although extension to other disaster settings would be ideal, we consider this beyond the scope of the current paper. Instead, we ask that you address limitations to generalization in your discussion section. However, we do expect that you will address all other requests for additional considerations and analyses this Reviewer makes (with particular emphasis on consumption), where the data for doing so are available.
- 2) Reviewer 2 raises quite substantial concerns about the lack of a causal theory of dependence in your study. We do not insist that you present causal evidence, as the primary goal of your study is prediction. However, please thoroughly revise to ensure that all interpretations and discussion of results accurately reflect the nature of your evidence, and that conclusions are not drawn that would require an underlying causal understanding.
- 3) In light of Reviewer 2's comments on the small R^2 values in your study, please ensure that your characterization of your results accurately reflects the effects shown.

If you wish to submit a suitably revised manuscript, we would hope to receive it within 4 months. I would be grateful if you could contact us as soon as possible if you foresee difficulties with meeting this target resubmission date.

- Include a "Response to the editors and reviewers" document detailing, point-by-point, how you addressed each editor and referee comment. If no action was taken to address a point, you must provide a compelling argument. When formatting this document, please respond to each reviewer comment individually, including the full text of the reviewer comment verbatim followed by your

response to the individual point. This response will be used by the editors to evaluate your revision and sent back to the reviewers along with the revised manuscript.

- Highlight all changes made to your manuscript or provide us with a version that tracks changes.

Link Redacted

Thank you for the opportunity to review your work. Please do not hesitate to contact me if you have any questions or would like to discuss the required revisions further.

Sincerely,

[REDACTED]

Reviewer expertise:

Reviewer #1: urban resilience, simulation methods

Reviewer #2: urban resilience, mobility data analysis

Reviewer #3: computational social science, urban mobility

REVIEWER COMMENTS:

Reviewer #1:

Remarks to the Author:

The study emphasizes the significance of understanding the dependencies between places, especially during shocks like pandemics. The perspective of dependence from human behavior is interesting, and this work contributes significantly to the field of economic resilience. While the manuscript offers innovative approaches to studying economic resilience, careful consideration of its methodological assumptions and the broader applicability of its findings is necessary.

1. The measurement for dependence of a POI i on another POI j (w_{ij}) only rely on the number of visits. However, other dimensions are ignored. For example, from a temporal perspective when a person visits office building A, and then visits restaurant B 30 minutes later, it would be a higher dependency compared to 90 minutes. In another other, the sequence also matters. Here are two cases: people go to places with the sequence of A, B, C should not be the same as A, C, B, when we talk about the dependency between A and B. I cannot access raw human mobility data to see if this can be modeled, but from a spatiotemporal perspective, I think this is still important.

2. The construction of dependency networks assumes that visits to different POIs within a short time frame imply economic dependence. This method simplifies complex economic relationships and may not capture indirect dependencies or the full breadth of economic interactions, such as online transactions or services not tied to physical visits.

3. In addition, an important aspect of economic resilience relies on its dependency in customer consumption. For example, if people spend money in A, and also spend money in B, we could say these are more dependent on each other economically. It is more important to see if they consume than if they visit the place. I understand not every POI is for customer spending. It makes sense to me that the authors use visit data to stand for dependency in Figure 1 and Figure 2. But when it comes to Figure 3 and Figure 4, the economic resilience discussion may not be operationalized with human mobility only. From Figure 3, the authors found that loss of visits to service and shopping stores has the most substantial correlation with other nodes through the behavior-based dependency network, which further justifies that place for consumption is more dependent on each other. I know that SafeGraph also offers spend information of POIs. I understand that there are data limitations with the spend information, but it is more related to economy.

4. The study's spatial granularity, while impressive, may still mask micro-level dynamics. For example, the analysis might not capture seasonal variations or the impact of temporary events (e.g., construction, holidays) on POI dependencies and urban economic resilience.

5. Using visitation change as the dependent variable, the authors demonstrated that using the behavior-based dependency network significantly improves the R^2 by 40% compared to using the distance-based null network. However, when we examine the resilience of business, the capability of recovery (also the recovery speed) to normal state is another important aspect. I wonder if behavior-based dependency still shows advantages. For example, if 95% of visits recovered for place A, does the behavior-based dependency make place B recover to a certain level.

6. A follow-up comment is it would be worth further discussing the correlation between visitation recovery of place A and the dependency network from neighbors. In the meantime, the question is how the authors define "neighbors" in Figure 3a is still unclear to me, since it is not in the physical dependency based on geographic distance, but a behavior dependency network. Maybe the authors mentioned it somewhere, but I missed it.

7. The authors use Leontief open model to model the cascading impacts. But the model itself assumes a linear impact. How could the authors address the non-linear cascading impacts? I think the COVID-19 may not affect the POIs in a linear way.

8. Moreover, even though COVID-19 is a good example, I think the visit reduction for POIs also highly relates to local intervention

policies, e.g., stay-at-home, gathering restrictions for restaurants, etc., not only the network dependency among POIs. I understand COVID-19 is a great example that applies to all cities, but if the authors could demonstrate the generalization to other shocks (e.g., urban disasters) in one or two cities (I know different cities have different natural disasters, so one or two case studies would be good enough), which may make the results more robust. I will leave this to the authors to decide.

9. Another concern for cascading impacts is that the temporal dimension is not modeled, potentially oversimplifying the resilience of urban economies to future shocks. For example, although the closure of college POIs could lead to a 5% visit decrease in both shopping and coffee POIs, if it takes less time for shopping POIs, we could still conclude that shopping POIs are more resilient. The chain effect over time is an important agenda for cascading impact analysis. It would be more interesting if Figure 4 b and c could extend to a temporal version, indicating how many days/weeks for the change to happen.

10. Also, the choice of parameters, such as a 50% visit reduction to all college POIs to simulate the shift to online education, is somewhat arbitrary. While it serves to illustrate the model's potential applications, the real-world scenarios could be far more complex, with varying degrees of impact across different sectors and times. More specifically, if it is possible to model the varying visit reductions to different college POIs?

Minor comments:

For figures like those showing behavior-based dependency networks, ensure that the visualization is not cluttered and that the differences between the sizes and colors of nodes are easily distinguishable. E.g., it is not easy to see the blue points in Figure 3a. The grey points in section 6.2 are also not easy to see.

In Figure 1, the visualization and interpretation of in-weights and out-weights for POIs could be complemented with a brief narrative on how these weights translate into real-world economic activities and urban dynamics.

Figure 1b and Figure 4c nodes are a little confusing to me. I understand each node is a subcategory for those main categories in Figure 1a. But we do not know what those subcategories are from Figure 1b, while some are labeled. The authors may consider offering more details in the supplementary file.

P. 10 in supplementary file, there is no alpha in the equation, which confused me.

Reviewer #2:

Remarks to the Author:

The authors propose a data-driven work of building point-of-interest (POI) co-visit networks using big mobility data to investigate urban economic resilience based on POI dependency.

The authors found that the proposed network could account for more long-distance dependencies than the gravity-based network. Accounting for POI dependency shows an improved predictability of economic resilience compared with only accounting for places' physical proximity.

The proposed study is similar to the existing work investigating population lifestyle based on a sequence of visiting multiple POIs using big mobility data. However, the paper offers a fresh perspective on discussing urban economic resilience based on POI dependency.

The authors did a fantastic job in data processing, analysis, and result visualization. Many existing works widely recognized the used data. The proposed measure (the number of occurrences divided by the number of total visits to one POI) to investigate the relationship between two POIs is simple (which is great) and elegant.

My biggest concern is the misinterpretation of the results. The paper lacks a theoretical foundation for POI dependency (or visiting one POI after another within a timeframe) and does not explore factors affecting POI dependency except for physical proximity. Therefore, the results of big data analysis only show the correlation between visiting one POI and visiting another one within a timeframe instead of causality (which is interpreted as dependency in the paper). Will the high correlation mean the causality here? Maybe yes, no, or perhaps partly explain the causality. However, we cannot draw a conclusion based on pure big data analysis without considering the theoretical factors (e.g., population, mobility option, infrastructure layout, distinguished needs for the essential businesses such as grocery and gas as well as non-essential businesses such as shopping malls and theaters).

Therefore, the inverse proposition, which is the key argument for urban economic resilience in the paper, is also problematic: failure of one POI will greatly affect another POI. Will failing to visit one POI lead to failing to visit another POI? The answer is undetermined without considering different factors affecting the relationships between two POIs, especially considering that the urban system is a highly complex system with much redundancy. It is more problematic when mixing the dependencies of essential POIs and Non-essential POIs (During COVID-19, the U.S. Cybersecurity and Infrastructure Security Agency (CISA) published guidance for essential business during the pandemic. At the same time, some states made adjustments based on this guidance.). Visits to essential POIs may not be affected much when the related POIs fail.

A widely used example in the paper is visiting POIs (e.g., restaurants or groceries) after visiting a university. The results of big data analysis show a high correlation between visiting these POIs. Indeed, the visits to restaurants may largely depend on the student population brought by the university, which is very common for many college towns in the US. The causality/factor here is the large student population brought by the university, affecting visits to other POIs. Understanding this will lead to different policies for urban economic resilience, such as maintaining the population level, keeping the proportion of the student population low, or increasing the income diversity of related POIs instead of preventing online education.

Another big concern is the presentation of the results. The authors highlight that accounting for POI dependency will improve predictability by 40% compared with distance-based models. However, the 40% here does not make sense as it is based on the adjusted R square. An adjusted R square will remain small if improved by 40% ($0.03 \times 1.4 = 0.042$). An improved R square remains small, as illustrated in Figure 3c, apparently less than 0.2 for LA. Then, the previous argument for the distance-based models may also be applied here: only explained limited variance. I would invite the authors to clearly label the adjusted R square for each

model in Figure 3c since the percentage improvement is misleading here.

A minor concern is that I only see one direction in Figure 1b. The dependence should be bi-directional according to the calculation. I may have some misunderstanding and would appreciate clarification here.

Using big mobility data, the paper offers a fresh perspective on urban economic resilience based on POI dependence. I appreciate the considerable workload and excellent data processing, analysis, and results visualization jobs. The paper is very well-written and well-organized. However, the paper lacks a theoretical foundation of POI dependence and does not explore factors affecting the dependence. The results of big data analysis are correlation instead of causality. Separating the essential POIs and non-essential POIs would be helpful. A deep conversation with a social scientist or urban economist will greatly help improve the paper.

Reviewer #3:

Remarks to the Author:

I would like to congratulate the authors for this paper. I think this is an excellent study, very well designed, and of great novelty. The use of behaviour-based dependency network is a very interesting idea, and, as the study demonstrates, a potentially very powerful one. I particularly like how this work emphasises the importance of behaviour over space, an issue which has been greatly overlooked so far. I believe this work will become a seminal study in this area.

The analysis is sound and rigorous, and a lot of validation against null models was carried out. The presentation of the manuscript is clear and well written.

I think this manuscript is ready for publication already, and I have no requests for the authors to do any changes.

Version 1:

Decision Letter:

8th April 2024

Dear Dr. Yabe,

RE: "Behavior-based dependency networks between places shape urban economic resilience"

Thank you for submitting your revised manuscript and for all your work on the revision.

Although your manuscript has been revised in response to reviewer comments, it does not fully comply with our editorial policies and formatting requirements. In particular, we require that all inferential statistical results be fully reported, including coefficient/effect size, exact p-value (for all $p > 0.001$), and confidence interval.

Before we can send the manuscript back to our reviewers, we ask that you revise it to ensure that it complies fully with our policies and is formatted according to our requirements. Specifically, we ask that you make two key changes:

1) Fully report all inferential statistical results discussed in the main text. In cases where you refer to the Supplementary Information for the underlying regression models, the main text (and/or figure caption) should include a reference to the specific table where each result can be found. Tables should include exact p-values and confidence intervals. Asterisks to denote statistical significance should not be used as a substitute for providing exact p-values.

2) We require that authors use an alpha level no larger than 0.05 for statistical tests, unless they preregistered the use of an alternative threshold. Please do not mark $p < 0.1$ as statistically significant.

To assist with this process, I have attached another copy of our checklist. If you have any questions regarding these points, please don't hesitate to contact me.

Link Redacted

Thank you in advance for attending to these requests and I look forward to receiving your revised manuscript.

Sincerely,
[REDACTED]

Version 2:

Decision Letter:

Our ref: NATHUMBEHAV-23114051B

13th August 2024

Dear Taka,

Thank you for submitting your revised manuscript "Behavior-based dependency networks between places shape urban economic resilience" (NATHUMBEHAV-23114051B). It has now been seen by the original referees and their comments are below. As you can see, the reviewers find that the paper has improved in revision. We will therefore be happy in principle to publish it in Nature Human Behaviour, pending minor revisions to comply with our editorial and formatting guidelines.

We are now performing detailed checks on your paper and will send you a checklist detailing our editorial and formatting requirements within three weeks. Please do not upload the final materials and make any revisions until you receive this additional information from us.

Sincerely,

[REDACTED]

Reviewer #1 (Remarks to the Author):

I appreciate the authors' careful revisions for the manuscript. My concerns have been fully addressed. I would like to congratulate the authors for this important work.

Reviewer #2 (Remarks to the Author):

The authors have addressed my concerns in the revised manuscript.

Reviewer #3 (Remarks to the Author):

As in my first review, I believe this to be a robust and strong manuscript, which has been improved further following the other reviews.

Version 3:

Decision Letter:

Dear Taka,

We are pleased to inform you that your Article "Behavior-based dependency networks between places shape urban economic resilience", has now been accepted for publication in Nature Human Behaviour.

Please note that *Nature Human Behaviour* is a Transformative Journal (TJ). Authors may publish their research with us through the traditional subscription access route or make their paper immediately open access through payment of an article-processing charge (APC). Authors will not be required to make a final decision about access to their article until it has been accepted. Find out more about Transformative Journals

Authors may need to take specific actions to achieve compliance with funder and institutional open access mandates. If your research is supported by a funder that requires immediate open access (e.g. according to Plan S principles) then you should select the gold OA route, and we will direct you to the compliant route where possible. For authors selecting the subscription publication route, the journal's standard licensing terms will need to be accepted, including self-archiving policies. Those licensing terms will supersede any other terms that the author or any third party may assert apply to any version of the manuscript.

With best regards,

[REDACTED]

P.S. Click on the following link if you would like to recommend Nature Human Behaviour to your librarian <http://www.nature.com/subscriptions/recommend.html#forms>

** Visit the Springer Nature Editorial and Publishing website at http://editorial-jobs.springernature.com?utm_source=ejp_NHumB_email&utm_medium=ejp_NHumB_email&utm_campaign=ejp_NHumB for more information about our career opportunities. If you have any questions please click [here](mailto:editorial.publishing.jobs@springernature.com).

Open Access This Peer Review File is licensed under a Creative Commons Attribution 4.0 International License, which permits use, sharing, adaptation, distribution and reproduction in any medium or format, as long as you give appropriate credit to the original author(s) and the source, provide a link to the Creative Commons license, and indicate if changes were made. In cases where reviewers are anonymous, credit should be given to 'Anonymous Referee' and the source. The images or other third party material in this Peer Review File are included in the article's Creative Commons license, unless indicated otherwise in a credit line to the material. If material is not included in the article's Creative Commons license and your intended use is not permitted by statutory regulation or exceeds the permitted use, you will need to obtain permission directly from the copyright holder.

Responses to the Reviewers for “Behavior-based dependency networks between places shape urban economic resilience”

Takahiro Yabe, Bernardo Garcia Bulle Bueno, Morgan Frank, Alex ‘Sandy’ Pentland, and Esteban Moro

We thank the reviewers for their helpful comments, questions, suggestions, and thorough review of our manuscript. We have extensively revised the manuscript based on the provided reviews to address their concerns. We list the revisions made and responses to each comment provided by the reviewers below.

Reviewer #1:

Remarks to the Author: The study emphasizes the significance of understanding the dependencies between places, especially during shocks like pandemics. The perspective of dependence from human behavior is interesting, and this work contributes significantly to the field of economic resilience. While the manuscript offers innovative approaches to studying economic resilience, careful consideration of its methodological assumptions and the broader applicability of its findings is necessary.

Thank you for the positive and constructive comments.

1. The measurement for dependence of a POI i on another POI j (w_{ij}) only rely on the number of visits. However, other dimensions are ignored. For example, from a temporal perspective when a person visits office building A, and then visits restaurant B 30 minutes later, it would be a higher dependency compared to 90 minutes. In another other, the sequence also matters. Here are two cases: people go to places with the sequence of A, B, C should not be the same as A, C, B, when we talk about the dependency between A and B. I cannot access raw human mobility data to see if this can be modeled, but from a spatiotemporal perspective, I think this is still important.

We thank the reviewer for this question and suggestion. In the Supplementary Material, we have performed extensive robustness checks on the impacts of parameter choice on the structure and predictive performance of the dependency networks.

We agree with your point on the effects of the time dimension on dependency weights. In Supplementary Note 2 and Figures S10-S14 (example for New York metropolitan area shown below in Figure R1), we find that the choice of temporal parameters (i.e., visitation interval time between places to be considered a ‘common visitor’) does not significantly affect the network properties, including the weight distribution, the relationship between average weight and physical distance between the POIs, and POI category dependency matrix, when the temporal threshold is larger than 1 hour. Furthermore, extensive experiments in Supplementary Note 5.3 shows the improvement in predictability of economic resilience is not affected by the choice of temporal threshold parameters.

Fig R1. Network characteristics under different maximum time differences for New York.

Moreover, to answer your question about the difference in A-B dependency between sequences $\{A, B, C\}$ and $\{A, C, B\}$, we tested whether the number of intermediate points (N_c) affects the above characteristics

of dependency networks. When $N_c=1$, only the $\{A, B, C\}$ sequence will count towards the A-B dependency, while when $N_c=2$, A-B dependency will be counted in both sequences $\{A, B, C\}$ and $\{A, C, B\}$. Through extensive robustness checks, as shown in Figures S10a-S14a, we found that the dependency network characteristics are not sensitive to the selection of the threshold value. An example for the New York metropolitan area shown below in Figure R2. This reflects the intuition that for some POI pairs, for example, gas stations and cafes, there is no necessity for one to precede the other. Furthermore, extensive experiments in Supplementary Note 5.3 shows the improvement in predictability of economic resilience is not affected by the choice of threshold parameters.

Fig R2. Network characteristics under different maximum step differences for New York.

We have revised our manuscript to highlight these points in the Results section (page 2):

“We demonstrate the robustness of the dependency network against the choice of the visit attribution parameters and the choice of the POI dataset in Supplementary Note 3. Furthermore, we show that the network characteristics are not sensitive to the choice of co-visit detection parameters, including the time interval between visits, and the number of intermediate stops in the sequence of visits.”

2. The construction of dependency networks assumes that visits to different POIs within a short time frame imply economic dependence. This method simplifies complex economic relationships and may not capture indirect dependencies or the full breadth of economic interactions, such as online transactions or services not tied to physical visits.

Thank you for your comment. We agree, our method measures the dependency relationships between places through the use of foot traffic patterns, and this may not capture the full breadth of economic interactions. As you rightly point out, we are not able to capture online transactions or services not tied to physical visits in this paper. The contribution of this paper is to introduce the concept of economic dependencies and estimation methods using mobility data. We acknowledge this limitation and added a discussion in the Discussion section:

“Four, this method measures the dependency relationships between places through foot traffic patterns, and this may not capture the full breadth of economic interactions. This approach simplifies complex economic relationships and may not capture other channels of economic interactions, such as online transactions or services not tied to physical visits. Follow up work using other behavior datasets, such as credit card purchase data would be an interesting avenue for future research [37].”

3. In addition, an important aspect of economic resilience relies on its dependency in customer consumption. For example, if people spend money in A, and also spend money in B, we could say these are more dependent on each other economically. It is more important to see if they consume than if they visit the place. I understand not every POI is for customer spending. It makes sense to me that the authors use visit data to stand for dependency in Figure 1 and Figure 2. But when it comes to Figure 3 and Figure 4, the economic resilience discussion may not be operationalized with human mobility only. From Figure 3, the authors found that loss of visits to service and shopping stores has the most substantial correlation

with other nodes through the behavior-based dependency network, which further justifies that place for consumption is more dependent on each other. I know that SafeGraph also offers spend information of POIs. I understand that there are data limitations with the spend information, but it is more related to economy.

Thank you for your comments and suggestions. We agree that using credit card transaction data would provide an additional dimension to our understanding of economic resilience. To address your point, we have investigated the use of Safegraph credit card data below. However, we would also like to point out that not all economic dependencies involve consumption activities. For example, restaurants may be highly dependent on offices and colleges, where consumption may not occur.

Following your suggestion, we have investigated the use of Safegraph spending data. Unfortunately, the credit card transaction data provided by Safegraph was not comprehensive, comprising around only 9.5% of the POIs present in the original dependency network dataset. This low coverage rate may contain biases towards certain business types and areas, which require additional data quality checks, including data representativeness analysis. Due to the low coverage and various uncertainties in the Safegraph credit card, we will avoid introducing this new dataset into the study, and leave this analysis for future work, using a more comprehensive spending dataset. This is reflected in the Discussion text, together with the previous comment:

“Four, this method measures the dependency relationships between places through foot traffic patterns, and this may not capture the full breadth of economic interactions. This approach simplifies complex economic relationships and may not capture other channels of economic interactions, such as online transactions or services not tied to physical visits. Follow up work using other behavior datasets, such as credit card purchase data would be an interesting avenue for future research [37].”

4. The study's spatial granularity, while impressive, may still mask micro-level dynamics. For example, the analysis might not capture seasonal variations or the impact of temporary events (e.g., construction, holidays) on POI dependencies and urban economic resilience.

Thank you for your comment. The goal of this work was to understand the general equilibrium dynamics of the dependency network. We agree that the equilibrium may be disrupted once in a while by events such as festivals, farmers markets, and construction, however these are temporary and rarely shift the equilibrium. We have further tested in Supplementary Note 3.1 that though there are seasonal variations, the general patterns and predictability of economic resilience is stable over time. Figures S18 and S19 show that the in- and out-weights across different time periods (2019 May – August and 2019 January – April) are highly correlated (Pearson correlation > 0.7) with the baseline time period (2019 September – December). Examples for New York, Boston, and Los Angeles areas shown below in Figure R3.

Fig R3. Comparison of in- and out-weights of each POI across different data collection periods for New York, Boston, and Seattle. In- and out-weights across different time periods (2019 May – August and 2019 January – April) are highly correlated ($\rho > 0.7$) with the baseline time period (2019 September – December).

Moreover, the category pairwise weight proportions were compared across different data collection periods, as shown in Figures S20 and S21. The category pairs that are highly dependent on each other are consistent across different time periods (2019 May – August and 2019 January – April) with the baseline time period (2019 September – December). Examples for New York, Boston, and Los Angeles areas shown below in Figure R4.

We have added explanation of these follow up experiments in the main text:

“The stability of dependency networks are further tested in Supplementary Note 3.1. Figures S18 -- S21 show that the in-weights, out-weights, and category pairwise weight proportions are highly correlated (Pearson correlation > 0.7) across different time periods (2019 May – August and 2019 January – April) with the baseline time period (2019 September – December).”

Fig R4. Comparison of category pairwise weight proportion across different data collection periods, for New York, Boston, and Seattle. The category pairs that are highly dependent on each other are consistent across different time periods (2019 May – August and 2019 January – April) with the baseline time period (2019 September – December).

5. Using visitation change as the dependent variable, the authors demonstrated that using the behavior-based dependency network significantly improves the R2 by 40% compared to using the distance-based null network. However, when we examine the resilience of business, the capability of recovery (also the recovery speed) to normal state is another important aspect. I wonder if behavior-based dependency still shows advantages. For example, if 95% of visits recovered for place A, does the behavior-based dependency make place B recover to a certain level.

Thank you for suggesting this interesting idea. Indeed, testing whether our method can better predict the recovery speed of POIs is of great interest. Figure R5 shows a schematic of a typical recovery curve of visitation patterns to a POI before, during, and after the pandemic. In the original experiment shown in Figure 3 and Supplementary Note 5, we tested the predictability of shock size using the following metrics: $\tilde{v}_i = v_i(t_1)/v_i(t_0) - 1$, $\tilde{v}_i = v_i(t_2)/v_i(t_0) - 1$, $\tilde{v}_i = v_i(t_3)/v_i(t_0) - 1$, which measured the size of visit reductions at different time points $t = t_1, t_2, t_3$, relative to visit rates before the pandemic at $t = t_0$.

We investigated whether we are able to predict the recovery speed better using the dependency network information using a similar method. Because the long-term recovery of visits may not necessarily recover back to pre-pandemic levels, we use two metrics to measure *recovery speed*: $\tilde{v}_i = v_i(t_2)/v_i(t_1) - 1$ and $\tilde{v}_i = v_i(t_3)/v_i(t_1) - 1$, which measures the magnitude of the recovery of visits with respect to the initial shock at time point $t = t_1$, which was during the initial pandemic period. Overall, consistent with previous results, we find that the predictability of visitation recovery significantly improves by using information

from the behavior-based dependency networks. Results for recovery at $t = t_2$ in New York, Boston, and Seattle (Tables S23, 24, 25) are shown below in Table R1 for reference.

We have added Supplementary Note Section 5.5, Tables S23 to S32, and the following text to the Supplementary Material:

“Tables S23 to S27 show the regression models for predicting the recovery of visits during 2020/6/1 and 2020/8/31, compared to the initial stages of the pandemic (2020/3/1 to 2020/5/31). Similarly, Tables S28 to S32 show the regression models for predicting the recovery of visits during 2020/9/1 and 2020/11/30, compared to the initial stages of the pandemic (2020/3/1 to 2020/5/31). In all metropolitan areas, the adjusted R^2 using the behavior-based dependency network effects (model 3) outperforms models 1 and 2 by 50% to 100%, showing substantial improvement in predictability. The regression coefficients for the behavior-based dependency network effects is significantly larger than the distance based effects, similar to the findings from the previous experiments. Overall the magnitude of the R^2 across all 4 models is smaller than those in the previous experiments, suggesting that the recovery dynamics is less predictable compared to visitation losses as we saw in the previous results in Section 5.4. Nevertheless, experiments using the recovery in visits during different time periods during the pandemic showed that the behavior-based dependency network effects substantially improve predictability compared to using the distance-based network effects.”

Moreover, we have added the following sentence in the main text:

“Additional analysis in Supplementary Note 5.5 showed that using the dependency network further improves the predictability of visitation recovery patterns (e.g., visitation during September - November 2020 compared to March - May 2020).”

Fig R5. Schematic showing a generic recovery trajectory of visits to POIs after the pandemic, and the different timings of the observations used to analyze the predictability of recovery.

Table S23: Linear regression models predicting the change in visits to POIs during the pandemic (2020 June - August) in New York compared to the initial stages of the pandemic (2020 March - May).

	Dependent variable: \bar{v}_i (Recovery of visits during the pandemic)			
	(1)	(2)	(3)	(4)
Constant	29.82 (26.09)	34.93 (25.16)	21.97 (25.16)	24.10 (25.14)
Distance effect $\sum_j \delta_{ij} \bar{v}_j$		19.23*** (0.429)		7.063*** (0.428)
Dependency effect $\sum_j w_{ij} \bar{v}_j$			56.12*** (0.447)	54.29*** (0.460)
Subcategory FE	Y	Y	Y	Y
PUMA FE	Y	Y	Y	Y
Observations	208,470	208,470	208,470	208,470
R^2	0.061	0.070	0.127	0.128
Adjusted R^2	0.060	0.069	0.126	0.127

Note: *p<0.1; **p<0.05; ***p<0.01

Table S24: Linear regression models predicting the change in visits to POIs during the pandemic (2020 June - August) in Boston compared to the initial stages of the pandemic (2020 March - May).

	Dependent variable: \bar{v}_i (Recovery of visits during the pandemic)			
	(1)	(2)	(3)	(4)
Constant	9.405 (43.92)	20.06 (42.78)	45.72 (42.78)	45.36 (42.78)
Distance effect $\sum_j \delta_{ij} \bar{v}_j$		25.71*** (1.259)		-2.29* (1.373)
Dependency effect $\sum_j w_{ij} \bar{v}_j$			64.42*** (1.274)	65.47*** (1.421)
Subcategory FE	Y	Y	Y	Y
PUMA FE	Y	Y	Y	Y
Observations	47,276	47,276	47,276	47,276
R^2	0.042	0.050	0.091	0.091
Adjusted R^2	0.039	0.048	0.089	0.089

Note: *p<0.1; **p<0.05; ***p<0.01

Table S25: Linear regression models predicting the change in visits to POIs during the pandemic (2020 June - August) in Seattle compared to the initial stages of the pandemic (2020 March - May).

	Dependent variable: \bar{v}_i (Recovery of visits during the pandemic)			
	(1)	(2)	(3)	(4)
Constant	11.98 (40.83)	13.94 (38.02)	35.70 (38.02)	35.42 (38.00)
Distance effect $\sum_j \delta_{ij} \bar{v}_j$		10.95*** (0.748)		-5.12*** (0.729)
Dependency effect $\sum_j w_{ij} \bar{v}_j$			59.46*** (0.759)	61.05*** (0.791)
Subcategory FE	Y	Y	Y	Y
PUMA FE	Y	Y	Y	Y
Observations	40,306	40,306	40,306	40,306
R^2	0.079	0.084	0.201	0.202
Adjusted R^2	0.077	0.081	0.199	0.200

Note: *p<0.1; **p<0.05; ***p<0.01

Table R1. Linear regression models predicting the change in visits to POIs during the pandemic (2020 June - August) compared to the initial stages of the pandemic (2020 March - May) in New York, Boston, and Seattle.

6. A follow-up comment is it would be worth further discussing the correlation between visitation recovery of place A and the dependency network from neighbors. In the meantime, the question is how the authors define "neighbors" in Figure 3a is still unclear to me, since it is not in the physical dependency based on geographic distance, but a behavior dependency network. Maybe the authors mentioned it somewhere, but I missed it.

Thank you for your question on the correlation between visitation recovery of place and the dependency network from neighbors. In fact, Figure 3b shows the Pearson Correlation between the changes in visits during the pandemic of the place of interest \tilde{v}_i and the aggregation of the visit changes of neighbors weighted by the dependency weights ($\sum_j w_{ij}\tilde{v}_j$). Neighbors of POI i are defined as nodes that have non-zero dependency weights with POI i , and are not based on the physical distance between POI i . We have defined these nodes as 'network neighbors', as opposed to neighbors based on physical proximity. We have added a sentence in page 5 to clarify this:

"Network neighbors j are defined as the set of nodes that have a non-zero dependency weight w_{ij} ."

7. The authors use Leontief open model to model the cascading impacts. But the model itself assumes a linear impact. How could the authors address the non-linear cascading impacts? I think the COVID-19 may not affect the POIs in a linear way.

Thank you for your suggestion. The contribution of this work is that we offer a novel approach to model economic spillovers. The goal of using the Leontief open model is to demonstrate its usefulness under simple (i.e., linear) assumptions first. Further improvements of the predictability using non-linear and complex models (e.g., graph neural networks) are beyond the scope of the current work, and we leave this for future work. We have added the following text to the limitation paragraph of the discussion:

"Third, in this study, simple linear models (e.g., Leontief open model) were used to test the effectiveness of our novel approach to modeling economic resilience. Given the advancements in non-linear and complex models for graph structured data (e.g., graph neural networks [42]), there is significant potential for future works that develop models that improve the predictability of resilience."

8. Moreover, even though COVID-19 is a good example, I think the visit reduction for POIs also highly relates to local intervention policies, e.g., stay-at-home, gathering restrictions for restaurants, etc., not only the network dependency among POIs. I understand COVID-19 is a great example that applies to all cities, but if the authors could demonstrate the generalization to other shocks (e.g., urban disasters) in one or two cities (I know different cities have different natural disasters, so one or two case studies would be good enough), which may make the results more robust. I will leave this to the authors to decide.

Thank you for your suggestion. We agree that it is worth demonstrating the generalization to other shocks, thus we have provided another example on college summer holidays in Supplementary Note 5.7. Although extension to other disaster settings would be ideal, we consider this beyond the scope of the current paper, but we agree that there are many potentially interesting shocks and natural disasters to investigate as future work. We highlight this potential for other applications of our method in the limitations section:

"The assumption in this study did not consider such dynamic reorganization because of the sudden and short-term nature of the shocks we analyzed (e.g., COVID-19, closures of colleges). Modeling the dynamics of the behavior-based dependency networks using human behavior data observed across a longer time frame, and applying the method to a broader range of realistic urban disruptions including climate change induced disasters could be an interesting topic for future research."

9. Another concern for cascading impacts is that the temporal dimension is not modeled, potentially oversimplifying the resilience of urban economies to future shocks. For example, although the closure of college POIs could lead to a 5% visit decrease in both shopping and coffee POIs, if it takes less time for shopping POIs, we could still conclude that shopping POIs are more resilient. The chain effect over time is an important agenda for cascading impact analysis. It would be more interesting if Figure 4 b and c could extend to a temporal version, indicating how many days/weeks for the change to happen.

Thank you for your suggestion. We have conducted extensive additional analysis on the recovery of visitation patterns during the pandemic, and showed that using the behavior-based dependency networks improves the predictability of recovery performance of POIs compared to using the distance based physical networks. Please refer to Supplementary Note Section 5.5, Tables S23 to S32, and our response to your comment #5 for more details on the predictability of visitation recovery. However, in our current model specification (Leontief open model), we cannot model the heterogeneity in the recovery speed of visitation patterns across places. To achieve this, we need to develop more complex models and simulations that are capable of describing the microscopic temporal dynamics of each place. We have added this discussion in the Discussion section:

“Third, in this study, simple linear models (e.g., Leontief open model) were used to test the effectiveness of our novel approach to modeling economic resilience. Given the advancements in non-linear and complex models for graph structured data (e.g., graph neural networks [47]), there is significant potential for future works that develop models that improve the predictability of resilience, and are capable of describing the microscopic temporal dynamics of disruption and recovery.”

10. Also, the choice of parameters, such as a 50% visit reduction to all college POIs to simulate the shift to online education, is somewhat arbitrary. While it serves to illustrate the model's potential applications, the real-world scenarios could be far more complex, with varying degrees of impact across different sectors and times. More specifically, if it is possible to model the varying visit reductions to different college POIs?

Thank you for your suggestion. Yes, the choice of “50%” is arbitrary. The method can be applied to any shock size of interest, and we provide alternative simulation shock sizes (25%, 100% visitation reduction to colleges) in Supplementary Note 6.2 and Figures S36 – S38. We have made the source code open so that simulations with any input parameters may be carried out on the infinite combinations of shock scenarios. Testing this method on various realistic shocks and disruption scenarios would be an interesting avenue for future research. We have added this discussion in the Discussion section:

“Modeling the dynamics of the behavior-based dependency networks using human behavior data observed across a longer time frame, and applying the method to a broader range of realistic urban disruptions including climate change induced disasters could be an interesting topic for future research.”

Minor comments:

1. For figures like those showing behavior-based dependency networks, ensure that the visualization is not cluttered and that the differences between the sizes and colors of nodes are easily distinguishable. E.g., it is not easy to see the blue points in Figure 3a. The grey points in section 6.2 are also not easy to see.

Thank you for your comments. There are much less blue points in Figure 3a (histogram shown in Figure S32) and it is inevitable that the blue points are less visible than red points. The objective of the Figure is to show the significant imbalance between the red and blue points, not the specific locations of where the blue points are located at. Similarly, the gray points in Figures S36 - S38 are meant to show the rough locations of the college POIs.

2. In Figure 1, the visualization and interpretation of in-weights and out-weights for POIs could be complemented with a brief narrative on how these weights translate into real-world economic activities and urban dynamics.

Thank you for your suggestion. We have added the following text to supplement our explanations of in-weights and out-weights:

“The total in-weight of each place measures to what extent the place is depended by other places in terms of customer visitation patterns, and the out-weight measures how much the place depends its customers on other places.”

3. Figure 1b and Figure 4c nodes are a little confusing to me. I understand each node is a subcategory for those main categories in Figure 1a. But we do not know what those subcategories are from Figure 1b, while some are labeled. The authors may consider offering more details in the supplementary file.

Thank you for your suggestion. We have added the original data file of the POI subcategory pairwise edge weights to the GitHub repository so that the readers can access the raw information.

4. P. 10 in supplementary file, there is no alpha in the equation, which confused me.

Thank you for pointing this out. We have fixed this in page 10 of the Supplementary material.

Reviewer #2:

Remarks to the Author:

The authors propose a data-driven work of building point-of-interest (POI) co-visit networks using big mobility data to investigate urban economic resilience based on POI dependency. The authors found that the proposed network could account for more long-distance dependencies than the gravity-based network. Accounting for POI dependency shows an improved predictability of economic resilience compared with only accounting for places' physical proximity. The proposed study is similar to the existing work investigating population lifestyle based on a sequence of visiting multiple POIs using big mobility data. However, the paper offers a fresh perspective on discussing urban economic resilience based on POI dependency. The authors did a fantastic job in data processing, analysis, and result visualization. Many existing works widely recognized the used data. The proposed measure (the number of occurrences divided by the number of total visits to one POI) to investigate the relationship between two POIs is simple (which is great) and elegant.

Thank you for the positive and constructive comments.

1. My biggest concern is the misinterpretation of the results. The paper lacks a theoretical foundation for POI dependency (or visiting one POI after another within a timeframe) and does not explore factors affecting POI dependency except for physical proximity. Therefore, the results of big data analysis only show the correlation between visiting one POI and visiting another one within a timeframe instead of causality (which is interpreted as dependency in the paper). Will the high correlation mean the causality here? Maybe yes, no, or perhaps partly explain the causality. However, we cannot draw a conclusion based on pure big data analysis without considering the theoretical factors (e.g., population, mobility option, infrastructure layout, distinguished needs for the essential businesses such as grocery and gas as well as non-essential businesses such as shopping malls and theaters).

Thank you for your comments. The primary goal of our study is prediction and not identifying causality from the data. The reviewer is correct in that we use the word “dependency” to describe the correlation between visiting one POI and visiting another POI within a timeframe. We have clarified this in the second paragraph in the Results section of the manuscript:

“Our dependency metric is the simplest way to encode the complex joint probability of visitation patterns to POIs in urban areas, however, it does not capture the causal mechanism of co-visitations.”

Even though our metric does not capture the causal mechanism, our dependency metric is highly predictive of future visitation patterns, see response to comment 2 below.

Moreover, distance is not the only factor we considered in our analysis of dependency between POI: in the regression models presented in Figure 2c and 2d in the main manuscript, we analyze the effects of physical distance on dependency weights conditioned on the POI category (as the reviewer suggest) and area where the POI is located (PUMA code, as a proxy for infrastructure layout) to provide more explanation of the POI dependencies. Please see Supplementary Note 4 for more details of the regression models.

We do agree that a more thorough discussion on the theoretical foundations of POI dependency could enrich the discussion and findings of the paper. To account for this point, we have added the following text to the Introduction paragraph:

“The household production theory, which holds that households allocate resources (time and money) to maximize utility, provides the theoretical basis for our study on patronage behavior across multiple stores

[22]. Empirical studies on consumer behavior have identified a multitude of factors that affect patronage to multiple stores, including customers' sociodemographic characteristics, store characteristics, available transportation modes, and the built environment [23-25]. Despite its importance on characterizing economic resilience, researchers have only recently started to examine general patterns of store patronage [26]. Due to lack of large-scale evidence of mobility and behavior patterns across stores, dependencies among businesses are typically measured by physical proximity, assuming similar patronage to nearby businesses. As a result, several studies have investigated the resilience of business areas based only on the type and diversity of businesses and amenities [27-29]. These studies fail to incorporate the actual patterns of how individuals visit and interact with different businesses and places.”

Moreover, in the Discussion section, we acknowledge that identifying causality was not the goal of the study, but it lays the groundwork for future investigations into causality:

“This study shows robust results on the predictability of economic resilience, laying the groundwork for future investigations into causal effects of dependencies on economic resilience (e.g., through natural experiments around natural disasters, rainfall, or construction projects that impact mobility).”

2. Therefore, the inverse proposition, which is the key argument for urban economic resilience in the paper, is also problematic: failure of one POI will greatly affect another POI. Will failing to visit one POI lead to failing to visit another POI? The answer is undetermined without considering different factors affecting the relationships between two POIs, especially considering that the urban system is a highly complex system with much redundancy. It is more problematic when mixing the dependencies of essential POIs and Non-essential POIs (During COVID-19, the U.S. Cybersecurity and Infrastructure Security Agency (CISA) published guidance for essential business during the pandemic. At the same time, some states made adjustments based on this guidance.). Visits to essential POIs may not be affected much when the related POIs fail.

Thank you for your comments. We believe the analysis in Figure 3 and the experiments conducted in Supplementary Note 5 answers your question. Figure 3c, Supplementary Figures S32-S35, and Tables S3-S42 show robust results showing that, even though the dependencies between POIs are not causal, the predictability of POI failure improves significantly by using information about the dependency relationships between POIs before the pandemic, and failures of neighboring POIs. To shed light on your question about the role of essential vs non-essential POIs on economic resilience, Figure 3b shows the heterogeneity in the category pairwise impacts of POI failures. The matrix shows that the failures of coffee, food, office, service, and shopping category POIs strongly affect other nodes (vertical patterns). In addition, as you suggest, we observe that health and office POIs (more essential) are less affected by failures of other nodes (horizontal pattern). We have added more explanation to discuss this point in the “Predictability of economic resilience via behavior-based dependency” section:

“We further observe that health and office POIs are least affected by the failure of alter nodes, suggesting structural differences in resilience of foot traffic among essential and non-essential POIs.”

3. A widely used example in the paper is visiting POIs (e.g., restaurants or groceries) after visiting a university. The results of big data analysis show a high correlation between visiting these POIs. Indeed, the visits to restaurants may largely depend on the student population brought by the university, which is very common for many college towns in the US. The causality/factor here is the large student population brought by the university, affecting visits to other POIs. Understanding this will lead to different policies for urban economic resilience, such as maintaining the population level, keeping the proportion of the student population low, or increasing the income diversity of related POIs instead of preventing online education.

Thank you for your interesting comments and insights. We are indeed able to break down the dependency weights across different sociodemographic groups, to identify which segments policies should target to maximize the impacts. In fact, a quick analysis shows that college campuses attract a much more diverse population than just students (suggested by the diversity in income of visitors), as shown in past research (Moro et al., 2021). Disaggregating the dependency network into different sociodemographic segments and

identifying policy targets will be an exciting avenue for future research. We discuss this point in the Discussion section:

“Moreover, following recent studies focusing on the substantial differences in mobility behavior across sociodemographic groups (e.g., [45,46]), we could decompose the dependencies into different income ranges, to better understand which sociodemographic segments are contributing to different types of dependency relationships.”

4. Another big concern is the presentation of the results. The authors highlight that accounting for POI dependency will improve predictability by 40% compared with distance-based models. However, the 40% here does not make sense as it is based on the adjusted R square. An adjusted R square will remain small if improved by 40% ($0.03 \times 1.4 = 0.042$). An improved R square remains small, as illustrated in Figure 3c, apparently less than 0.2 for LA. Then, the previous argument for the distance-based models may also be applied here: only explained limited variance. I would invite the authors to clearly label the adjusted R square for each model in Figure 3c since the percentage improvement is misleading here.

We appreciate your suggestion. While we agree that the absolute adjusted R² values may not be very high, we argue that this is common in social science and economics research due to the high complexity of socioeconomic processes, and that the substantial improvement above the baseline method is the contribution of this paper. To present the results clearly, we have added the adjusted R² values in the text and the subtitles of Figure 3. Now it reads:

“Using the behavior-based dependency network significantly improves the adjusted R^2 by 40% compared to using the distance-based null network (from 0.17 to 0.24).”

5. A minor concern is that I only see one direction in Figure 1b. The dependence should be bi-directional according to the calculation. I may have some misunderstanding and would appreciate clarification here.

Thank you, we have added the bi-directionality to the visualization in Figure 1b.

Using big mobility data, the paper offers a fresh perspective on urban economic resilience based on POI dependence. I appreciate the considerable workload and excellent data processing, analysis, and results visualization jobs. The paper is very well-written and well-organized. However, the paper lacks a theoretical foundation of POI dependence and does not explore factors affecting the dependence. The results of big data analysis are correlation instead of causality. Separating the essential POIs and non-essential POIs would be helpful. A deep conversation with a social scientist or urban economist will greatly help improve the paper.

We appreciate your positive and constructive comments! We think we were able to substantially improve the theoretical aspects of the paper.

Reviewer #3:

Remarks to the Author:

I would like to congratulate the authors for this paper. I think this is an excellent study, very well designed, and of great novelty. The use of behaviour-based dependency network is a very interesting idea, and, as the study demonstrates, a potentially very powerful one. I particularly like how this work emphasises the importance of behaviour over space, an issue which has been greatly overlooked so far. I believe this work will become a seminal study in this area. The analysis is sound and rigorous, and a lot of validation against null models was carried out. The presentation of the manuscript is clear and well written. I think this manuscript is ready for publication already, and I have no requests for the authors to do any changes.

Thank you very much for your extremely positive and encouraging comments!